# Verification of anthropogenic VOC emission inventory through ambient measurements and satellite retrievals

Jing Li[1,2], Yufang Hao[1], Maimaiti Simayi[1], Yuqi Shi[1], Ziyan Xi[1], and Shaodong Xie[1]

[1]College of Environmental Sciences and Engineering, State Key Joint Laboratory of Environmental Simulation and Pollution Control, Peking University, Beijing, 100871, PR China
[2]Department of Environmental Health, Harvard T.H. Chan School of Public Health, Boston, 02215, USA

*Correspondence to:* Shaodong Xie (xiesd@pku.edu.cn)

**Abstract.** Improving the accuracy of the anthropogenic volatile organic compound (VOC) emission inventory is essential for reducing air pollution. In this study, we established an emission inventory of anthropogenic VOCs in the Beijing–Tianjin–Hebei (BTH) region of China for 2015 based on the emission factor (EF) method. Online ambient VOC observations were conducted in one urban area of Beijing in January, April, July, and October, which respectively represented winter, spring, summer, and autumn in 2015. Furthermore, the developed emission inventory was evaluated by a comprehensive verification system based on the measurements and satellite retrieval results. Firstly, emissions of the individual species of the emission inventory were evaluated according to the ambient measurements and emission ratios versus carbon monoxide (CO). Secondly, the source structure of the emission inventory was evaluated using source appointment with the Positive Matrix Factorization (PMF) model. Thirdly, the spatial and temporal distribution of the developed emission inventory was evaluated by a satellite-derived emission inventory. According to the results of the emission inventory, the total anthropogenic VOC emissions in the BTH region were 3277.66 Gg in 2015. Online measurements showed that the average mixing ratio of VOCs in Beijing was approximately 49.94 ppbv in 2015, ranging from 10.67 ppbv to 245.54 ppbv. The annual emissions for 51 of 56 kinds of non-methane hydrocarbon species derived from the measurements were agreed within ±100% with the results of the emission inventory. Based on the PMF results and the emission inventory, it is evident that vehicle-related emissions dominate the composition of anthropogenic VOCs in Beijing. The spatial correlation between the emission inventory and satellite inversion result was significant (p < 0.01) with a correlation coefficient of 0.75. However, there were discrepancies between the relative contributions of fuel combustion, emissions of oxygenated VOCs (OVOCs), and halocarbons from the measurements and inventory. To obtain a more accurate emission inventory, we propose the investigation of the household coal consumption, the adjustment of EFs based on the latest pollution control policies, and the verification of the source profiles of OVOCs and halocarbons.

## 1 Introduction

Ambient volatile organic compounds (VOCs) encompass of various kinds of chemical substances that predominantly help to form the ground-level ozone ($O_3$) and the secondary organic aerosol (SOA) (Toro et al., 2006). Their direct emission sources include biogenic sources, forest fires, and anthropogenic sources (Guenther et al., 2006;Kansal, 2009;Simpson et al., 2011). Most VOCs are emitted naturally on a global scale; however, in urbanized areas, they are mainly emitted anthropogenically (Guenther et al., 2012;Janssens-Maenhout et al., 2015).

In addition, some VOC species have adverse effects on human health (Bari et al., 2016;Weichenthal et al., 2012). Therefore, it is essential to acquire reliable knowledge of anthropogenic VOC emissions to develop strategies for reducing the emissions and production of secondary pollutants.

The emission inventory is a widely used method for calculating the emissions of VOCs, through which the magnitude, strength, spatial and temporal distribution, source structure of VOC emissions, and other related information can be provided (Wang et al., 2014). Moreover, emission inventories are essential input data for the chemical transport model (Hodzic et al., 2010;Coll et al., 2010). Since the pioneering study by (Piccot et al., 1992), the "Emission factor (EF) method" has been widely used to establish emission inventories, which estimates the

emissions of VOC sources by multiplying the corresponding activities and detailed EFs (Tonooka et al., 2001;Streets, 2003;Klimont et al., 2002;Ohara et al., 2007;Bo et al., 2008;Zhang et al., 2009;Zheng et al., 2009;Li et al., 2014;Wu et al., 2016a;Huang et al., 2017;Janssens-Maenhout et al., 2015). The characterization and quantification of VOC emissions are highly complicated as the emission sources of VOCs exhibit complexity and diversity (Borbon et al., 2013). Although great progress has been made in the establishment of VOC emission

inventories, there are still a number of limitations (Wu et al., 2016a). Based on the results of the Monte Carlo simulation, the uncertainty of the VOC emission inventory was more than 100% (Bo et al., 2008). The chemical transport model simulation studies found that existing VOC emissions inventories cannot accurately assess air quality, which makes it ineffective for meeting the management needs of developing emission reduction policies (Carmichael, 2003;Coll et al., 2010;Kim et al., 2011). Thus, developing an accurate emission inventory of

anthropogenic. VOCs is key to effectively control and reduce air pollution in the future.

        Emission inventories can be verified and evaluated based on the ambient measurements of VOCs or satellite retrievals. However, the concentration of VOCs is measured after the emission undergoes physical and chemical transformations. One method for assessing regional emissions is to perform source appointments with receptor models that can calculate different sources' contributions and assess the VOC source structures, in terms of the

accuracy of the emission inventories, accordingly (Gaimoz et al., 2011;Morino et al., 2011;Wang et al., 2014). In addition, the calculation of the anthropogenic emission of individual VOC species can be performed according to two indicators, (1) the emission ratios to an inert tracer (reference compound) and (2) the known emissions for the inert tracer (Borbon et al., 2013). Subsequently, the results can be used for the verification of the species-specific emissions of the EF-based inventory. Since the satellite data possess the advantage of reflecting the spatial

characteristics of VOCs (Fu et al., 2007), satellite-derived anthropogenic VOC emission estimations obtained from the chemical transport model can be utilized to evaluate the spatial distribution of the EF based emission inventories (Cao et al., 2018).

        Earlier studies by various research groups applied only one of these methods to evaluate either the source structure or species-specific emissions of VOC emission inventories (Gaimoz et al., 2011;Borbon et al.,

2013;Wang et al., 2014;Cao et al., 2018). Moreover, most studies have been based on the data from one or two-month ambient measurements, which cannot accurately represent the annual emissions (Gaimoz et al., 2011;Borbon et al., 2013;Wang et al., 2014). Also, there is a lack of systematic analysis of the qualities and uncertainties of anthropogenic VOC emission inventories. Therefore, we took the VOC emissions of the Beijing–Tianjin–Hebei (BTH) region as a case for the verification of a method for establishing an anthropogenic VOC

emission inventory. The BTH region is the political center of China, a developed cluster of large cities, and one of the most polluted areas in China (Jiang et al., 2015). In recent years, severe haze events have occurred frequently

in the BTH region (Zhu et al., 2016), and the mixing ratios of $O_3$ have increased significantly (Ma et al., 2016). There is an urgent need for effective pollution control measures in the area.

In this study, the emission inventory of anthropogenic VOCs in 2015 in the BTH region of China was developed at a 3 km × 3 km spatial resolution based on the EF method. We conducted online measurements of ambient VOCs at an interval of 1 h in January, April, July, and October 2015 at an urban site in Beijing. We extracted the VOC emissions of the BTH region from a satellite-derived anthropogenic VOC emission inventory. The source structure, species-specific VOC emissions, and spatial-temporal distribution of the emission inventory were evaluated based on online measurements and satellite retrievals.

## 2 Methodology

### 2.1 Establishment of the anthropogenic VOC emission inventory

Through a systematic literature review, we found that the anthropogenic VOC emission inventory methodologies are similar, but the source classification and EFs used in some studies are different. So, this study combined the existing source classification system, EF databases, and source profile databases, followed by establishing the emission inventory in the BTH region (Bo et al., 2008;Wu et al., 2016a;Huang et al., 2017;Zhao et al., 2017;Zhong et al., 2017). A detailed description of the method is provided below.

### 2.1.1 Source classification

According to the actual state of VOC emissions in the BTH region, a four-level categorization was used to classify the anthropogenic VOC sources (Wu et al., 2016a). Level 1 has five sub-levels in total: transportation, the stationary combustion of fossil fuel, biomass burning, solvent utilization, and industrial processes. For example, transportation, a Level 1 source, was further divided into off-road transportation and on-road vehicles in Level 2. On-road vehicles can be divided into buses, passenger cars, motorcycles, light-duty vehicles, as well as heavy-duty vehicles in Level 3 based on the fleet type. Level 3 sources can be further divided into Level 4 sources based on fuels. Table S1 lists these classifications in detail.

### 2.1.2 Emission estimation and allocation

Calculation of on-road vehicular VOC emissions was performed taking into account the EFs, number of vehicles, and the corresponding average mileage for each vehicle category per year, following Eq. (1).

$$E_V = \sum P_{i,j} \times VMT_{i,j} \times EF_{i,j} \qquad \text{Eq. (1)}$$

where $E_v$ is the vehicular VOC emission (Gg); $P_{i,j}$ denotes the number of vehicles in category $i$ under emission standard $j$; $VMT_{i,j}$ denotes the average mileage per year in km for vehicles in category $i$ under emission standard $j$; and $EF_{i,j}$ denotes the emission factor in g km$^{-1}$ for vehicles in category $i$ under emission standard $j$. EFs were calculated by COmputer Programme to calculate Emissions from Road Transport version 4 (COPERT 4), a widely used software application for calculating emissions from road transport. The input parameters included the fuel information, monthly ambient temperature, average speeds and so on. The method has been explained in detail in previous studies (Cai and Xie, 2013). Furthermore, the vehicles were classified into different categories according to COPERT 4.

The Level 2 sources of biomass burning include biofuel combustion and crop field residue burning. Multiplying the activity data by corresponding EFs yields the emissions estimate of biomass burning. For crop residue burning in fields, the activity data was the total mass of crop residues burned in fields, which can be calculated based on the crop production, the grain-to-straw ratio of each crop type, the proportion of field crop residue burning, the burning efficiency, and the proportion of dry matter in the crop residue. Li et al. (2016) described in detail the method used to obtain the activity data. The EFs of biomass burning were obtained from local experimentally measurements, as shown in Table S1.

The emission estimation of other VOC sources, including off-road transportation, the stationary combustion of fossil fuel, industrial processes, and solvent utilization, were estimated by multiplying the corresponding activities and detailed EFs. The EFs used in this study were obtained from locally measured EFs, recently published literature, national or local discharge standards, the EPA AP42 Report (*http://www3.epa.gov/ttnchie1/ap42/*) and the EEA Air Pollutant Emission Inventory Guidebook 2013. The detailed sub-sources and the EFs of each sub-source are listed in Table S1.

In the current study, selected county-level statistical data were prioritized to calculate the VOC emissions. For the sources without county-level data, either city-level or provincial-level statistical data were selected (Table S2). Further, the VOC emissions estimated based on the statistical data were allocated to county-level by the most related surrogates. GDP, population, and cultivation area were used as the surrogates for allocating industrial sources, residential sources, and biomass burning sources, respectively. The county-level emissions were further divided into 3 km ×3 km grids based on a population density map (*http://www.geodoi.ac.cn/WebCn/Default.aspx*). For field crop residue burning, the allocation of emissions was carried out according to the fire counts in croplands. The MODIS Thermal Anomalies/Fire gridded level-3 product (MOD/MYD14A1) was selected to determine fire counts. The CCI-LC Map was used to identify croplands (*http://maps.elie.ucl.ac.be/CCI/viewer/index.php*). Multiplying the total VOC emission by the corresponding weight percentage from the source profile database yielded the emission of individual VOC species. The source profile database used in this study was listed in Table S4. The study by Wu and Xie (2017) described the source profile database used in this study in detail.

The monthly variability of this VOC emission inventory was calculated based on the monthly profiles (Table S3). In summary, monthly profiles for industrial emissions were developed based on the monthly output of industrial products (NBS, 2015). Power plant monthly profile was derived from monthly statistics of power generation (NBS, 2015). Monthly profiles of residential fossil fuel combustion were estimated based on household survey results (Guo et al., 2015;Zheng et al., 2014). Monthly profile of on-road vehicle emissions was derived from Li et al. (2017b). For field crop residue burning, the monthly profile was estimated based on the MODIS fire counts in croplands (Li et al., 2016). We assumed that there was no monthly variation for the emissions from the other sources (Wu and Xie, 2018).

**2.2 VOC sampling and analysis**

The online observations of ambient VOCs were conducted at an interval of 1 h in January, April, July, and October 2015, representing winter, spring, summer, and fall, respectively. The roof of the Technical Physics Building at Peking University, with a height of approximately 15 m above the ground (PKU, 39.99 °N, 116.33 °E, Fig. 1) was selected as the sampling site. During this study, a total of 2174 valid measurements were obtained. No large industrial point sources are around to the monitoring site. This site has been used to represent a typical urban

environment in Beijing in many studies (Song et al., 2007;Yuan et al., 2012;Li et al., 2015;Wang et al., 2015;Wu et al., 2016b).

The sampling and analysis method of this study follow the US EPA Method TO-15 (USEPA, 1999). A custom-built online system was used to collect and analyze the ambient VOCs in a continuous and automatic

manner (*TH-PKU 300B, Wuhan Tianhong Instrument Co. Ltd., China*). The system is a gas chromatography-mass spectrometry/flame ionization detector (GC-MS/FID) with a time resolution of 1 hour (GCMS-QP2010, Shimadzu Co., Ltd., Japan). A total of 104 C2-C11 VOC species belonging to alkanes (27), alkenes (13), aromatics (16), halocarbons (29), alkynes (1), nitriles (1), and oxygenated VOCs (OVOCs, 17) were recognized and quantified by standard gases (*source from the Environmental Technology Center, Canada, and Linde Electronics and*

*Specialty Gases Inc., USA*). These standard gases are ideal for use with the TO-15 Calibration Standards (Linde, 2017). In addition, five concentrations of standard gases were used to perform monthly calibrations. The method detection limit (MDL) exhibited by the GC-MS/FID was in the range of 0.002 ppbv to 0.070 ppbv for each targeted species. A more detailed description of this system has been provided elsewhere (Li et al., 2015;Li et al., 2018).

**2.3 Emission inventory verification system**

The validation of the emission inventory was conducted based on ambient VOC measurements and satellite retrieval results. Firstly, emission strengths of VOCs on the basis of emission ratios relative to CO were estimated, and the results were compared with the emission inventory developed here to evaluate the species-specific emissions. Secondly, the positive matrix factorization (PMF) receptor model was applied to evaluate the VOC source structure with regard to its accuracy in the emission inventory. Thirdly, the study compared the emission

inventory developed here with a satellite-based anthropogenic VOC emission inventory to evaluate the spatial distribution and annual VOC emissions.

**2.3.1 Species VOC emissions based on ambient measurements**

VOC mixing ratios obtained from field observations cannot be directly compared with the VOC emissions due to physical and chemical transformation processes. One widely used approach to compare them is to estimate the

175 emissions of individual VOC species by their emission ratios to a reference compound and the known emissions for the reference compound (Fu et al., 2007;Hsu et al., 2010;Shao et al., 2011;Borbon et al., 2013;Wang et al., 2014). The theory of this method is that the relative ratios between enhancements of VOCs and the increasing of a trace gas could reflect the ratios of their emission strength (Shao et al., 2011). The relative ratios can help reduce the influence of physical transformation processes. The calculation of annual emissions for individual VOC

species is performed based on Eq. (2):

$$E_{VOC} = E_{Ref} \times ER_{VOC} \times MW_{VOC}/MW_{Ref} \qquad \qquad \text{Eq. (2)}$$

where, $E_{VOC}$ denotes the emission of a particular VOC species per year (Gg); $E_{Ref}$ denotes the emission of the reference compound per year (Gg); $MW_{VOC}$ denotes the molecular weight of a particular VOC species; $MW_{Ref}$ denotes the molecular weight of the reference compound; and $ER_{VOC}$ denotes the emission ratio of VOC species

relative to the reference compound (ppbv (ppmv Ref)$^{-1}$).

Photochemical processing is an important factor influencing the chemical compositions of VOCs in ambient air. Thus, the way in which photochemical processing impacts the measured VOC ratios should be excluded or corrected using a temporal filter (Borbon et al., 2013). The local time period 03:00 to 07:00 was set as a temporal

filter to reduce the impact of photochemical processing (Wang et al., 2014). The emission ratios of VOC species to the reference compound $(ER_{VOC})$ from 03:00 to 07:00 local time were estimated using the linear fit model.

In this study, we selected CO as a reference compound considering that: (1) CO has similar sources as that of anthropogenic VOC and (2) CO emissions show lower uncertainty compared with VOC emissions (Warneke et al., 2007;Wang et al., 2014). Thus, CO was a suitable reference compound (Coll et al., 2010;Borbon et al., 2013;Wang et al., 2014). The CO levels in the ambient air were obtained from the Wanliu National Air Quality Monitoring Station (*http://zx.bjmemc.com.cn*). The annual emission value of CO was obtained from the CO emission inventory of the MarcoPolo Project (*http://www.marcopolo-panda.eu*, (Hooyberghs et al., 2016)), which was copied from the Multi-resolution Emission Inventory for China (MEIC) emissions (http://www.meicmodel.org/index.html). This emission inventory has been validated by the chemical transport model (Hu et al., 2017), satellite observations (Yumimoto et al., 2014), and comparison with other studies (Li et al., 2017a). The spatial resolution of this CO emission inventory is $0.25\,°\times0.25\,°$. To obtain reasonable results, we compared the VOC emissions of the grid where the PKU site was located (Fig. S1).

This approach for the calculation of VOC emissions based on ambient observations has several limitations. First, in this study, we evaluated the emission inventory based on VOC measurement at one site, which limits the spatial representation of VOC measurement data relative to those observations in more sites. Secondly, we assume that the air mass over the site could respect the average emissions of the grid box, which will lead some uncertainties. Thirdly, these approach relies on the assumption that the composition of urban emissions relative to CO. Thus, emissions based on VOC measurements on multiply multiple sampling sites would be more reliable and some other method such as chemical transport model simulation may be an ideal approach to verify emission inventories based on field observations in our future study.

**2.3.2 Source apportionments**

The U.S.EPA PMF 5.0 model (USEPA, 2014b) was applied to the ambient VOC source apportionments. More information about the PMF model can be seen elsewhere (Paatero and Tapper, 1994), while the section below gives a brief discussion on some related concepts that enable a better understanding of the analysis in this study. Since the PMF model is a mathematical approach to quantify the contribution of sources, it is necessary to use a large number of samples to ensure the reliability of the results. During the collection period, a total of 2174 valid measurements were obtained for 104 VOC species. The PMF required two input files: (1) concentration values of sample species and (2) the uncertainty values of sample species. The observed uncertainty file was set following the method proposed by Polissar et al. (1998), which was recommended by the user guide of the PMF model. The uncertainty is calculated by Eq. (3), if the mixing ratio is equal to or less than the MDL; the uncertainty is calculated using Eq. (4), if mixing ratio if larger than the MDL (USEPA, 2014b).

$$\text{Uncertainty} = \frac{5}{6} \times MDL \qquad\qquad Eq.\,(3)$$

$$\text{Uncertainty} = \sqrt{(Error\ Fraction\ \times mixing\ ratio)^2 + (0.5\ \times MDL)^2} \qquad\qquad Eq.\,(4)$$

The PMF model can calculate each species' signal to noise ratio (S/N) according to the input files. Species were categorized as "Bad" when the S/N ratio was < 0.2 and "Weak" when the S/N ratio was > 0.2 but < 0.5. Two to ten factors solutions were calculated by the PMF model. The most appropriate number of factors were selected by some mathematical indicators calculated following the PMF model, including the coefficient of determination,

Q value, a possible explanation of the sources, and the residual distribution.

### 2.3.3 Satellite-derived emission inventory

To evaluate the spatial distribution of the EF based VOC emission inventory and the VOC annual emissions, we compared the emission inventory established in our study with satellite-derived anthropogenic VOC emission estimations. The satellite-derived estimations were obtained from the Global Emission project (*http://tropo.aeronomie.be/datapage_BIRA.php?species=TNMVOC*). The anthropogenic VOC emissions were derived based on source inversion with the IMAGESv2 model (Stavrakou et al., 2009) which is constrained by the column densities of tropospheric HCHO from OMI satellite instrument (De Smedt et al., 2015). HCHO is a high-yield product of many VOC species oxidation (Millet et al., 2006;Stavrakou et al., 2015). Its atmospheric lifetime is relatively short (only a few hours) against photolysis and oxidation. Its column concentration is directly related to the emission of reactive VOCs (De Smedt et al., 2015). Therefore, satellite observations of formaldehyde column concentrations can provide a "top-down" constraint for better quantitation of high spatial-temporal resolution of VOCs. The spatial resolution of the satellite-derived emissions was $0.25\,° \times 0.25\,°$ and the temporal resolution was one month. To facilitate the spatial distribution verification, the emissions of the 3 km $\times$ 3 km emission inventory established in this study were weighted and summed in the $0.25\,° \times 0.25\,°$ grid, so that the spatial resolution was consistent with the satellite inversion emission inventory.

### 3 Results and discussion

### 3.1 VOC emission inventory in the BTH region in 2015

A total of 3277.66 Gg of anthropogenic VOC were emitted in BTH region in 2015, accounting for about 10% of the national VOC emissions (Wu et al., 2016a). Emissions in the Beijing, Tianjin, and Hebei provinces were 411.72 Gg, 666.53 Gg, and 2199.41 Gg, respectively. The spatial distribution of VOC emissions is shown in Fig. 2. The emissions were lower in the northern part of the BTH region, and the emission density was higher in the southeast. Moreover, the southwestern part of Beijing, the southeastern part of Tianjin, and the eastern part of Shijiazhuang displayed high VOC emissions.

Figure 3 illustrates the contribution of each source to the total VOC emissions. For the BTH region, industrial processes were the largest source, accounting for 39% of the total VOC emissions. The next was transportation with emissions of 1080.62 Gg, accounting for 33% of the total emissions. Emissions from solvent utilization, biomass burning, and fuel combustion contributed 18%, 6%, and 4 % of total VOC emissions, respectively. The primary source of VOC emissions in Beijing was transportation, while in the Tianjin and Hebei provinces it was industrial processes.

Figure 4 illustrates the chemical compositions of VOC emissions in the BTH region. Emissions of a total of 152 VOC species (Table S4) bellowing to alkanes, alkenes, alkynes, aromatics, halocarbons, OVOCs, nitriles, and others were calculated in this study. The emissions of aromatics, alkanes, OVOCs, and alkenes accounted for 34%, 32%, 17%, and 11% of total anthropogenic VOC emissions, respectively. Aromatics and alkanes were the main compound groups of anthropogenic VOCs, with annual emissions of 1412.6 Gg and 1058.6 Gg, respectively. Emissions of alkynes, halocarbons, other VOCs and nitrile were much lower, accounting for 2.2%, 2.2%, 1.4%, and 0.2% of total anthropogenic VOC emissions, respectively. The species with the highest emissions are listed

in Fig. 4. The top ten species are m/p-xylene, toluene, ethylbenzene, ethylene, n-hexane, benzene, ethane, ethanol, o-xylene, and isopentane with annual emissions of 227.0 Gg, 216.9 Gg, 154.5 Gg, 142.1 Gg, 140.8 Gg, 138.1 Gg, 113.3 Gg, 102.2 Gg, 94.7 Gg and 91.1 Gg, respectively. Among the 10 species, five belonged to aromatics, three to alkanes, one to ethene, and one to OVOCs.

**3.2 Verification of species-specific VOC emissions**

**3.2.1 VOC mixing ratios**

Figure 5 presents the average mixing ratios and chemical compositions of VOCs measured at the PKU site in January, April, July, and October 2015. The average mixing ratio of VOCs was about 49.94 ppbv in 2015, varying from 10.67 ppbv to 245.54 ppbv. The highest VOC mixing ratio was occurred in January, with an average value of 62.26 ppbv. The VOC mixing ratios were relatively lower in April and July, with average values of 41.09 ppbv and 41.77 ppbv, respectively. In October, the average mixing ratio of VOCs was 50.64 ppbv. The VOCs species varied dramatically across the four seasons. Overall, alkanes dominated total VOCs during all seasons, accounting for 31.2%−39.5% on average. In January, alkanes constituted the largest group of VOCs (39.5%), and the next one was alkenes (22.6%). In other months, the highest VOC group was also alkanes, followed by OVOCs. Detailed time series and box-plot of VOC mixing ratios observed in this study are shown in Fig. S2 and Fig. S4, respectively.

The VOC species showing the highest mixing ratios in January, April, July, and October (top 20) are listed in Table 1. Ethane exhibited the largest proportion during all the four months. Ethene, acetylene, propene, and benzene are considered typical combustion tracers (Liu et al., 2008). Compared with the other months, the mixing ratios of combustion sources tracers were much higher in January. Benzene and toluene were important VOC species. During combustion processing, the emissions of benzene are much higher than toluene (Barletta et al., 2005). As shown in Table 1, benzene showed a higher mixing ratio than toluene in January. In contrast, the benzene showed a lower mixing ratio compared with toluene in the other three months. Combustion may be an important source in winter. During July, the levels of acetone, methyl methacrylate, and 2-butanone were much higher, which may be influenced by secondary formation process. The mixing ratios of n-butane and i-pentane were also very high in July, which may be influenced by the evaporation of gasoline in summer.

**3.2.2 Verification of individual VOC species emissions**

Here, we selected CO as a reference compound for anthropogenic VOC species. For verifying the rationale of setting CO as the reference compound, this study analyzed the mutual influence of mixing ratios of individual VOC species and CO levels. All VOC species, except β-pinene and $C_2F_3Cl_3$, were significantly related to CO ($p < 0.05$). Acetylene, ethane, and ethene were the most obviously related to CO ($R > 0.8$), and benzene, propene, tran-2- butane, 2,3-dimethyl butane, and i-butane also displayed strong correlations with CO ($R > 0.6$). The correlation coefficients between some halocarbons/ketones and CO were lower as halocarbons and ketones have a few emission sources different from CO sources. Affected by the biogenic emissions (Guenther et al., 2006), isoprene had a stronger correlation with CO in winter (R=0.77), and had a weaker correlation with CO in summer (R=0.18).

Table S5 listed the emission ratios for individual VOC species measured in this study and compared the

individual emissions estimated from the emission ratios and emission inventory. Based on the annual emissions derived from the measurements, the top fifteen species with the highest emissions were ethane, ethene, propane, acetylene, acetone, toluene, dichloromethane, n-butane, benzene, methyl methacrylate, ethyl acetate, propene, i-pentane, i-butane, and 2-butanone. Based on the annual emissions derived from the emission inventory, the top fifteen species with higher emissions were m-/p-xylene, toluene, ethylbenzene, benzene, ethylene, o-xylene, i-pentane, ethane, tetrachloroethylene, ethyl acetate, 1,2, 4-trimethylbenzene, propylene, pentane, n-butane, and n-hexane.

The emissions of individual VOC species determined by the measurements and emission inventory for the sampling site in 2015 were displayed in Fig. 6. After the comparison with results obtained from measurements, the emissions for 51 of 56 kinds of non-methane hydrocarbon (NMHC) species were agreed within ±100% in the emission inventory, 15 species agreed within ±50%, and 10 species agreed within ±25%. The acetonitrile emissions determined by the measurements is around five times the acetonitrile emissions determined by the emission inventory. The annual emissions of many OVOCs and halocarbons were much lower in the emission inventory than that in the measurements results. Moreover, the emission levels of some aromatics in the emission inventory were higher than those of the measurements.

Figure 7 makes a direct comparison between the emissions of individual VOC species from the measurements and emission inventory. The annual emissions for alkanes were agreed within ±100% between the two methods, except ethane and propane, which are important tracers of natural gas and LPG (Katzenstein et al., 2003). The emission inventory may underestimate the VOC emissions from the utilization of natural gas and LPG. This conclusion was consistent with the study by Wang et al. (2014), which compared the emission ratios of 27 sites in Beijing to the INTEX-B emission inventory.

The annual emissions for the alkenes, except ethene, were agreed within ±100%. The acetylene in the emission inventory showed a lower annual emission compared with that from the measurements. Ethene and acetylene are mainly emitted through an incomplete combustion process (Liu et al., 2008;Mo et al., 2016). The lower emissions of the two species in the emission inventory indicated that the emission inventory might have underestimated the VOC emissions from combustion sources. Emissions of some aromatics like toluene, o-xylene, and m/p-xylene in the emission inventory were higher compared with that from the measurements. Toluene and xylene were mainly emitted from various solvent utilization sources, such as automobile coatings, printing, and furniture manufacturing. The common character of these sources is that Beijing had issued local VOC emission standards for the above sources since 2015 (DB11/1201-2015; DB11/1202-2015; DB11/1226-2015; DB11/1227-2015; BD11/1228-2015). The enactment of local VOC emission standards decreases the emission of some aromatics. However, the influence of those standards on EFs has not been considered before, which might lead to higher emissions of toluene, o-xylene, and m/p-xylene in the emission inventory than in measurements.

OVOC and halocarbons showed a much lower emission in the emission inventory compared with that from the observations, which might be due to the lack of reliable source profiles. In addition, secondary production through the oxidation of precursor VOCs might impact the accuracy of OVOC emission estimated from the observations. The VOC source profiles obtained from the local measurements mainly focused on the monitoring of NMHC species, and fewer studies have been conducted on the measurement of OVOCs and halocarbons (Mo et al., 2016). Moreover, the extensively used VOC source profiles database, USEPA SPECIATE, excludes VOC species with very low atmospheric photochemical reactivity, such as methylene chloride, methyl chloroform, and

fluorochemicals (USEPA, 2014a). For instance, both acetone and 2-butanone are carbonyl compounds. Vehicle exhaust, biomass combustion, food flue gas, surface coating, and industrial exhaust gas are the sources of carbonyl compounds, while only 45 of the more than 150 sources profiles included acetone and 38 source profiles contained 2-butanone. Another example is that of esters, which are used as tracers of industrial emissions and are emitted

by most of the industrial processes and surface coating processes. However, only 12 sources in the source profile database contained methyl acetate, 6 sources contained methyl methacrylate, 29 sources contained ethyl acetate, and 14 sources contained butyl acetate. Halocarbons are also vital tracers of industrial sources. Of all the source profile databases collected in this study, only 33 sources profiles contained methyl chloride, 28 sources contained chloroform, 35 sources contained 1,2-dichloroethane, and 26 sources contained 1,2-dichloropropane.

Consequently, more source emission monitoring investigations need to be carried out to obtain reliable source profiles of VOC emissions.

### 3.3 Verification of source structures

### 3.3.1 Source apportionments

The PMF receptor model was used to conduct dynamic source apportionment according to the measured VOCs.

The following analysis adopted 63 strong and 5 weak species, accounting for more than 90% of the total VOC mixing ratios. In the following, six sources were identified, including (1) vehicle-related sources, (2) fuel combustion, (3) aged air mass and biomass burning, (4) industrial processes, (5) biogenic source, (6) solvent utilization. Figure S5 shows the source profiles of individual sources, and the source identification is described in the supplemental file.

Source contribution percentages in January, April, July, and October was shown in Table S6. In January, fuel combustion made the most significant contribution (55%) to VOC mixing ratios. Emissions from vehicle–related source and industrial processes contributed 19%, and 14% to total VOC mixing ratios, respectively. The contributions from aged air mass, solvent utilization, and biogenic were relatively low at 7%, 3%, and 1%, respectively. In April, aged air mass made the largest contribution (33%) to VOCs, followed by vehicle-related

sources (22%) and industrial processes (21%).The contribution proportion of fuel combustion, solvent utilization, and biogenic respectively was 12%, 7%, and 5% of the total VOC mixing ratios. In July, vehicle-related sources accounted for 50% of the total VOCs. Meanwhile, the biogenic source elevated from 5% in April to 18% in July. Emissions from solvent utilization, aged air mass, industrial processes, and fuel combustion respectively contributed 12%, 10%, 6%, and 4% of total VOCs. In October, vehicle-related sources were also the most

important source, accounting for 33% of the total VOC mixing ratios, followed by solvent utilization (23%), aged air mass (18%), and industrial processes (16%). Contributions from fuel combustion and biogenic sources were relatively low, with values of 6% and 5%, respectively.

Comparison of the relative contributions of VOC emission sources in Beijing calculated by the PMF model of this study and results from the other studies during these seasons was listed in Table S7. Results of this and

other studies have shown that the fuel combustion was the largest VOC contributor in winter. The contribution proportions of fuel combustion in winter were ranged from 45% - 55% (Li et al., 2015;Yang et al., 2018). Results of this and other studies have shown that vehicle-related source was the largest VOC contributor in summer and winter, with the contribution ranged from 50% - 57%, and 33% - 42%, respectively. The contribution proportion

of summer biogenic emission in this study was larger than that in the other studies.

**3.3.2 Comparison between the emission inventory and PMF results**

Figure 8 illustrates the comparison between source profiles derived from the PMF against their attributed sources from the emission inventory. The source profiles for fuel combustion agreed between the two methods. For the source profiles of transportation, the contributions of C2 − C4 alkanes derived from the PMF were larger than contributions from the emission inventory. Aromatics were the dominant group in the source profiles for solvent utilization derived from the two methods. For the source profiles of industrial processes, the proportions of some halocarbons OVOCs from the PMF were larger than proportions from the emission inventory. The finding agreed with the results of section 3.2.2 that the proportions of OVOCs and halocarbons in the source profile database may be unreliable.

This study compared the annual average and seasonal PMF results with the VOC source structures of the emission inventory established (Fig. S6 and Fig. 9). The annual results of PMF demonstrated that transportation made the largest contribution to VOCs. On the other hand, the results of the emission inventory also showed that transportation was the largest contributor to VOCs. The contribution of industrial processes from PMF results was comparable with the result from emission inventory. The contribution of the solvent utilization obtained from the PMF result was higher than the value of the emission inventory. The relative contribution of the fuel combustion source from the PMF result was significantly lower than that in the emission inventory (8%). According to the PMF results and the emission inventory, the vehicle-related emissions were the primary source in urban area in Beijing. Solvent utilization and industrial processes also contributed significantly. However, large differences were observed in the stationary combustion of fossil fuel consistent with the comparison results shown in section 3.2.2.

Compared with the seasonal PMF results, the emissions from industrial processes, transportation, and solvent utilization of the emission inventory didn't exhibit obvious seasonal variations (Fig. 9). It is because the monthly profiles of these sources, which developed on monthly statistics, have little monthly variations (Wu and Xie, 2018). The emissions from fuel combustion of the emission inventory exhibit similar seasonal variations with the PMF results with much higher emissions in winter than the other seasons. However, the relative contribution of fuel combustion for each season in the emission inventory was significantly lower than the contribution in the PMF results, especially for winter.

On the basis of the above comparisons, we inferred that: (1) the annual contributions of the vehicles, solvent utilization, and industrial processes from the emission inventory and the PMF results were similar, but the monthly profiles of these sources cannot replicate the temporal variations; and (2) the fuel combustion in the emission inventory showed a considerably lower relative contribution than the value from the PMF analysis, especially in winter, the central heating season in Beijing. The emission inventory was established by multiplying statistical data and EFs. Table S1 lists the EFs of fossil fuel combustion sources used here, which were referenced from local measurement studies and were comparable with EFs reported by the US government or the European Union (USEPA, 1995;EEA, 2013). Hence, we attributed the large difference between fuel combustion contributions from the PMF analysis and emission inventory to the uncertainties of activity data obtained from statistical information. For industrial-related activity data, the statistical data was relatively reliable; however, uncertainty in the statistical data was very high for residential-related activity data. For instance, it is challenging to quantify the consumption

of coal briquettes and chunks, the major fossil fuel for heating and cooking in Chinese households, since such coal for household often goes unreported in official statistics (Liu et al., 2016). Moreover, it is suspected that actual residential coal consumption is much higher than it reported in official statistics (Andersson et al., 2015). We speculate that the underestimation of the emissions from stationary combustion of fossil fuel by the emission inventory can be explained by the incomplete statistical data of residential coal consumption. In future studies, it is necessary to estimate the exact activity data and emissions of residential fossil fuel combustion through scientific approaches.

### 3.4 Verification of spatial and temporal distributions

The satellite-derived emission inventory revealed that the VOC emissions in the BTH region were 4368.50 Gg, while the annual emissions according to the EF-based emission inventory were 3277.66 Gg. The deviation of the emissions calculated by the two methods is 30%, within a reasonable error range. The two types of emission inventory shown similar distributions (Fig. 10a) and comparable emissions (Fig. 10b) for the gridded emissions. The spatial distributions of VOC emissions derived from the emission inventory and satellite data are shown in Fig. 11. The gridded VOC emissions of the emission inventory displayed significant correlations with the emissions derived from the satellite ($p < 0.05$), with a correlation coefficient of 0.75. The areas with higher and lower VOC emissions in the two emission inventories are also consistent. The areas with higher emissions were concentrated in the urban areas of Beijing and Tianjin, whereas areas with lower emissions were concentrated in the north of the BTH region, including Zhangjiakou and Chengde. The emissions of EF-based emission inventory present some positive biases compared to the satellite-derived emissions for the grids located in the south (Fig. 10b). Emissions calculated by the EF-based emission inventory were higher than those calculated by the satellite-derived emission inventory for Xingtai and Handan (Fig. 11), two heavy industrial cities in Hebei province.

The temporal resolution of the satellite-derived emission inventory is one month. As shown in Fig. 12, monthly variations of VOC emissions exhibit obvious seasonal characteristics, with a maximum in winter and a minimum in summer, which are consistent with the seasonal characteristics of the ambient VOC mixing ratios (Fig. 5). However, monthly profiles for the EF-based emission inventory, which developed based on monthly statistics, didn't exhibit seasonal variations. EF-based VOC emission inventories developed by the other studies (Li et al., 2017b;Wu and Xie, 2018) also didn't exhibit obvious seasonal variations because of little monthly variation in emissions from transportation, industrial processes, and solvent utilization (Wu and Xie, 2018). The satellite-derived emission inventories may better reflect the monthly characteristics of VOC emissions and be used to allocate monthly emissions.

The satellite-derived emission inventory possesses the advantage of efficiently reflecting the spatial and monthly characteristics of the VOC emissions (Palmer et al., 2006;Fu et al., 2007;Marais et al., 2014;Stavrakou et al., 2015;Bauwens et al., 2016). The significant correlation between the emission inventory established here and the satellite-derived emission inventory is indicative of the reliability of the spatial allocation method and the spatial emissions of the emission inventory. The satellite-derived emission inventory can provide constrains to improve the existing understanding of monthly profiles of VOC emissions.

**4 Conclusions**

An emission inventory of anthropogenic VOCs was established for the BTH region in China at a 3 km × 3 km spatial resolution based on the EF method. We conducted VOC online observations selecting a site in the urban area of Beijing. Based on the measurements, we estimated the annual emission strengths of VOCs according to their emission ratios relative to CO, and then compared these results with the emission inventory established in this study to verify the species-specific VOC emissions. The PMF model was used to qualify the relative

contribution made by each source. The result was compared with the emission inventory to evaluate the source structure of the VOCs. We also compared the emission inventory established in our study with a satellite-derived anthropogenic VOC emission inventory for verification of the spatial distribution and VOCs annual emissions.

According to the PMF results and the emission inventory, the vehicle-related emissions dominate the composition of anthropogenic VOCs in Beijing. The annual emissions of 91% NMHCs derived from the

measurements were agreed within ± 100% with the results of the emission inventory. The total amount of anthropogenic VOC emissions of the emission inventory was similar to the satellite inversion result, with a deviation of 30%. The spatial correlation between the emission inventory and the satellite inversion result was significant ($p < 0.01$) with a correlation coefficient of 0.75.

Our results showed that the vehicle-related VOC emissions estimated by the emission inventory based on the

EF method were reliable. The emissions of NMHCs estimated by the emission inventory were accurate, and the method of spatial distribution was feasible. Nevertheless, there are a few limitations of the existing method for establishing an anthropogenic VOC emission inventory based on EFs. Firstly, there is a large difference between the relative contributions of fuel combustion, and the method underestimated the emissions from fuel combustion sources, especially in winter. Secondly, the emissions of OVOCs and halocarbons estimated in the emission

inventory appeared much lower than those derived from the measurements due to the lack of reliable source profiles. Thirdly, emissions of some aromatics estimated in the emission inventory were higher than the values derived from the measurements due to the enactment of some local emission standards. Fourthly, monthly profiles of the EF-based emission inventory developed based on monthly statistics cannot replicate the temporal variations of VOC emissions well.

To acquire a more accurate VOC emission inventory, we propose the following improvements to the future emission inventories: (1) the investigation of household coal consumption to improve the accuracy of activity data; (2) the adjustment of EFs based on the latest pollution control policies to establish dynamic emissions inventories; and (3) the verification of OVOCs and halocarbons emissions and (4) the performance of more source emission monitoring studies to obtain reliable source profiles of VOC emissions.

**Author contributions.**

SDX designed the study, JL performed the data analysis and wrote the paper. YFH contributed to the online measurements. MS contributed to the development of the emission inventory. YQS and ZYX assisted with data collection. All authors assisted with interpretation of the results and the writing of the paper.

**Competing interests.**

The authors declare that they have no conflict of interest.

**Acknowledgements.**

This work was supported by the National Natural Science Foundation of China for "The development and validation of emission inventories of anthropogenic volatile organic compounds in the Beijing–Tianjin–Hebei region, China" (Grants No. 91544106), and by the National Air Pollution Prevention Joint Research Center of 495    China for "The research of characteristics, emission reduction and regulatory system of volatile organic compounds in key sectors" (Grants No. DQGG0204).

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

**Figure and Table captions:**

Table 1. The VOC species showing the highest mixing ratios (ppbv) in January, April, July, and October 2015 (top 20).

Figure 1. The location of the BTH region in China (shaded area) and the sampling site (black dot).

Figure 2. The spatial distribution of VOC emissions in the BTH region at a 3 km $\times$ 3 km resolution.

Figure 3. Source contributions to the total VOC emission in the Beijing, Tianjin, Hebei provinces (a) and the BTH region (b), China.

Figure 4. The top 30 VOC species with the highest emissions (bar plot) and the contributions of VOC groups to the total emsssions (pie plot)VOC emissions in the BTH region, China.

Figure 5. Mixing ratios and compositions of VOCs (ppbv) measured at the PKU site during the four seasons and throughout the study period.

Figure 6. Comparisons of VOC annual emissions (ton) derived from the ambient measures and emission inventory for the PKU site ( $0.25 °\times 0.25 °$grid).

Figure 7. Comparisons of individual species emissions (ton) derived from the ambient measures and emission inventory for the PKU site ( $0.25 °\times 0.25 °$grid).

Figure 8.Source profiles for VOCs calculated by PMF and the emission inventory (Species name was shown in Table S4).

Figure 9.Monthly VOC source structure identified by the PMF analysis (left) and the emission inventory (right).

Figure 10. Comparisons between the emissions from the EF-derived emission inventory and the satellite-derived emission inventory.

Figure 11. Comparision between spatial distribution of VOC emissions obtained from the EF-derived emission inventory and the satellite-derived emission inventory with a $0.25 °\times 0.25 °$spatial resolution.

Figure 12. Monthly variability of VOC emissions from the satellite-derived and EF-based emission inventories.

**Table 1. The VOC species showing the highest mixing ratios (ppbv) in January, April, July, and October 2015 (top 20).**

| January | | April | | July | | October | |
|---|---|---|---|---|---|---|---|
| Ethane | 11.66 | Ethane | 5.67 | Ethane | 4.18 | Ethane | 5.63 |
| Ethene | 10.70 | Acetone | 5.53 | Acetone | 4.18 | Propane | 4.54 |
| Acetylene | 6.98 | Propane | 3.49 | Methylmethacrylate | 3.75 | Acetone | 4.42 |
| Propane | 5.48 | Ethene | 2.58 | Propane | 2.98 | Ethene | 3.73 |
| CF2Cl2 | 2.58 | Acetylene | 2.41 | Acetylene | 2.47 | Acetylene | 3.09 |
| Propene | 2.45 | Dichloromethane | 2.34 | Dichloromethane | 2.04 | Dichloromethane | 2.41 |
| n-Butane | 2.10 | n-Butane | 1.29 | Ethene | 1.82 | n-Butane | 1.95 |
| Acetone | 1.82 | Ethyl acetate | 1.13 | Toluene | 1.55 | Methylmethacrylate | 1.89 |
| i-Butane | 1.45 | Toluene | 1.06 | n-Butane | 1.54 | Toluene | 1.66 |
| Benzene | 1.30 | 2-Butanone | 1.04 | 2-Butanone | 1.00 | Chloroform | 1.40 |
| Toluene | 1.28 | Chloromethane | 1.03 | i-Penpane | 1.00 | Chloromethane | 1.40 |
| i-Penpane | 1.13 | i-Penpane | 0.88 | i-Butane | 0.86 | i-Penpane | 1.39 |
| Dichloromethane | 1.12 | i-Butane | 0.87 | Benzene | 0.80 | i-Butane | 1.10 |
| 2-Butanone | 0.95 | methylmethacrylate | 0.85 | 1,2- Dichloroethane | 0.78 | Ethyl acetate | 1.03 |
| Ethyl acetate | 0.89 | Chloroform | 0.78 | Chloromethane | 0.76 | Benzene | 1.01 |
| Chloromethane | 0.81 | Benzene | 0.76 | Chloroform | 0.66 | Propene | 0.87 |
| Penpane | 0.68 | Penpane | 0.59 | Acetonitrile | 0.61 | Penpane | 0.82 |
| Methylmethacrylate | 0.53 | Propene | 0.57 | 1,1,2,2-Detrachloromethane | 0.57 | n-Hexane | 0.74 |
| Butyl acetate | 0.49 | Acetonitrile | 0.50 | Penpane | 0.50 | 1,2-Dichloropropane | 0.60 |
| n-Hexane | 0.44 | Menthyl acetate | 0.48 | 1,2-Dichloropropan | 0.49 | 1,2-Dichloroethane | 0.59 |

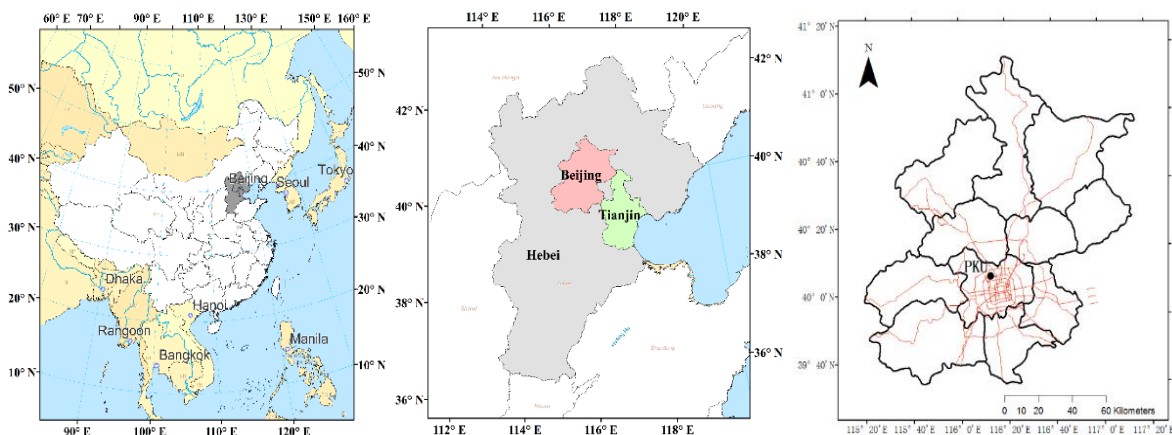

**Figure 1. The location of the BTH region in China (shaded area) and the sampling site (black dot).**

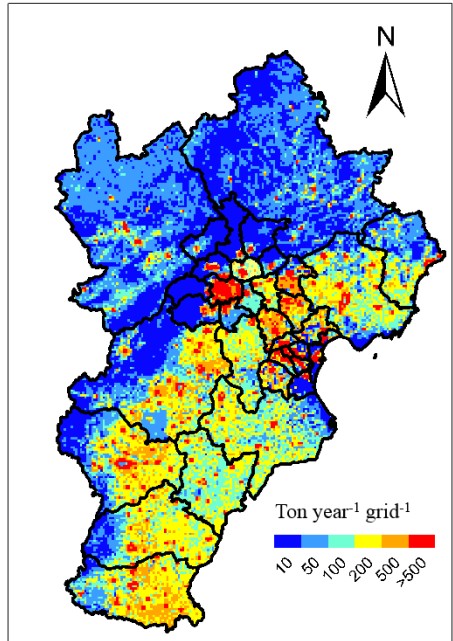

**Figure 2. The spatial distribution of VOC emissions in the BTH region at a 3 km × 3 km resolution.**

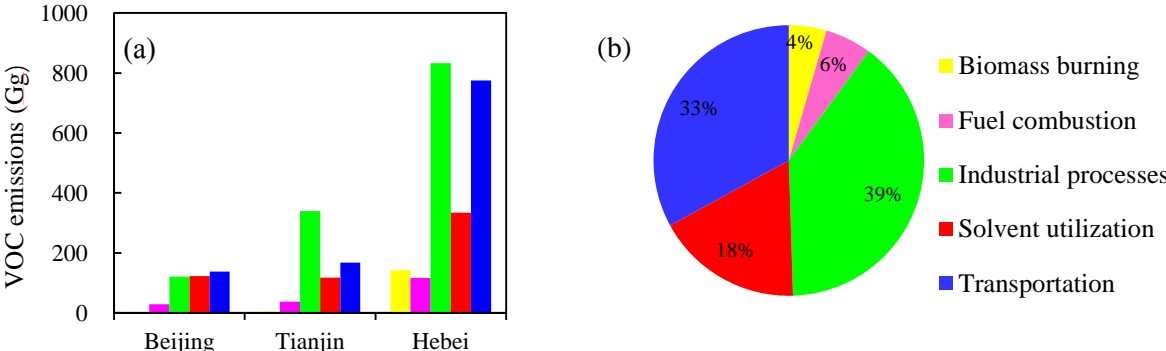

**Figure 3. Source contributions to the total VOC emission in the Beijing, Tianjin, Hebei provinces (a) and the BTH region (b), China.**

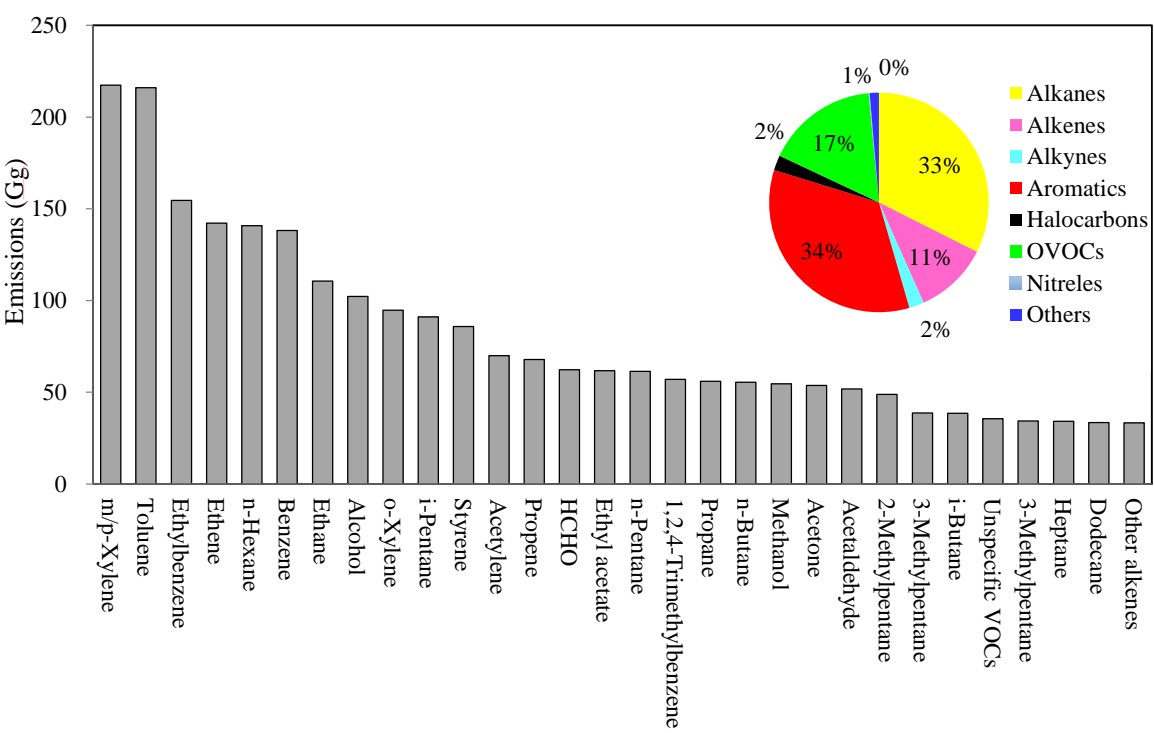

**Figure 4. The top 30 VOC species with the highest emissions (bar plot) and the contributions of VOC groups to the total emsssions (pie plot) in the BTH region, China.**

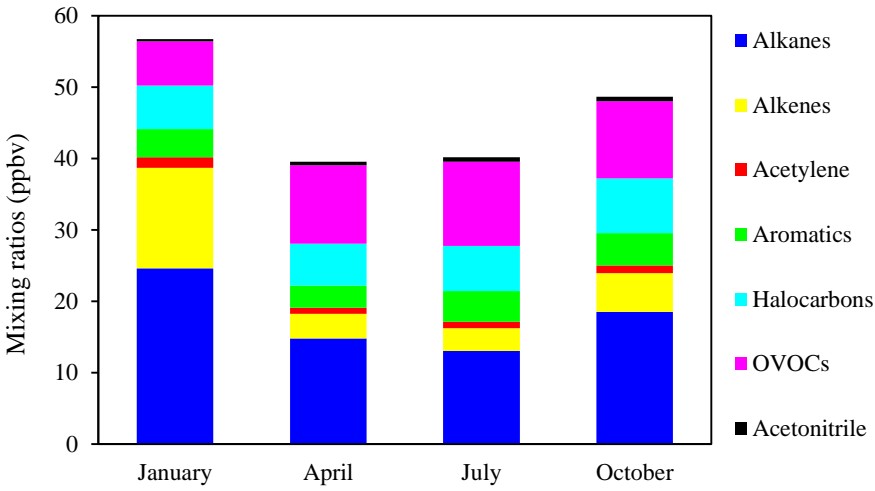

**Figure 5. Mixing ratios and compositions of VOCs (ppbv) measured at the PKU site during the four seasons and throughtout the entire study period.**

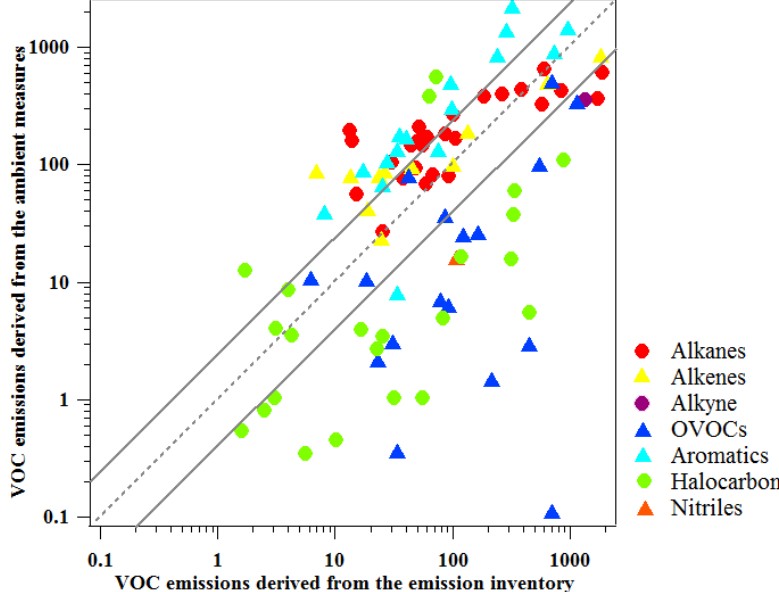

**Figure 6. Comparisons of VOC annual emissions (ton) derived from the ambient measures and emission inventory for the PKU site ( 0.25 ° × 0.25 ° grid).**

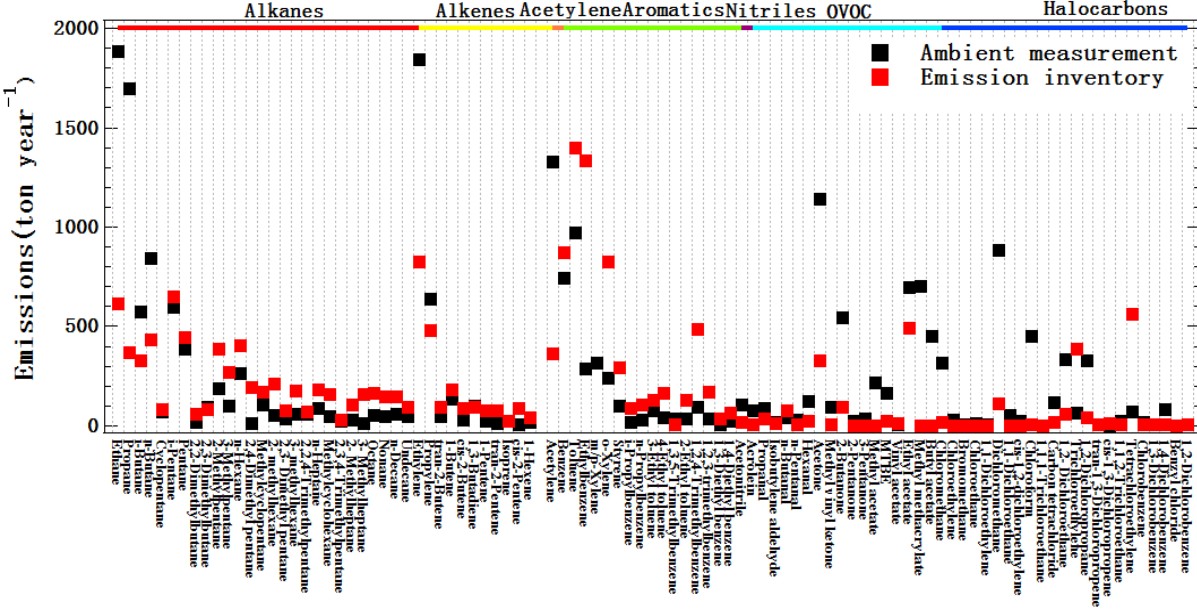

**Figure 7. Comparisons of individual species emissions (ton) derived from the ambient measures and emission inventory for the PKU site ( 0.25 ° × 0.25 °grid).**

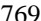

Species ID

**Figure 8. Source profiles for VOCs calculated by PMF and the emission inventory (Species name was shown in Table S4).**

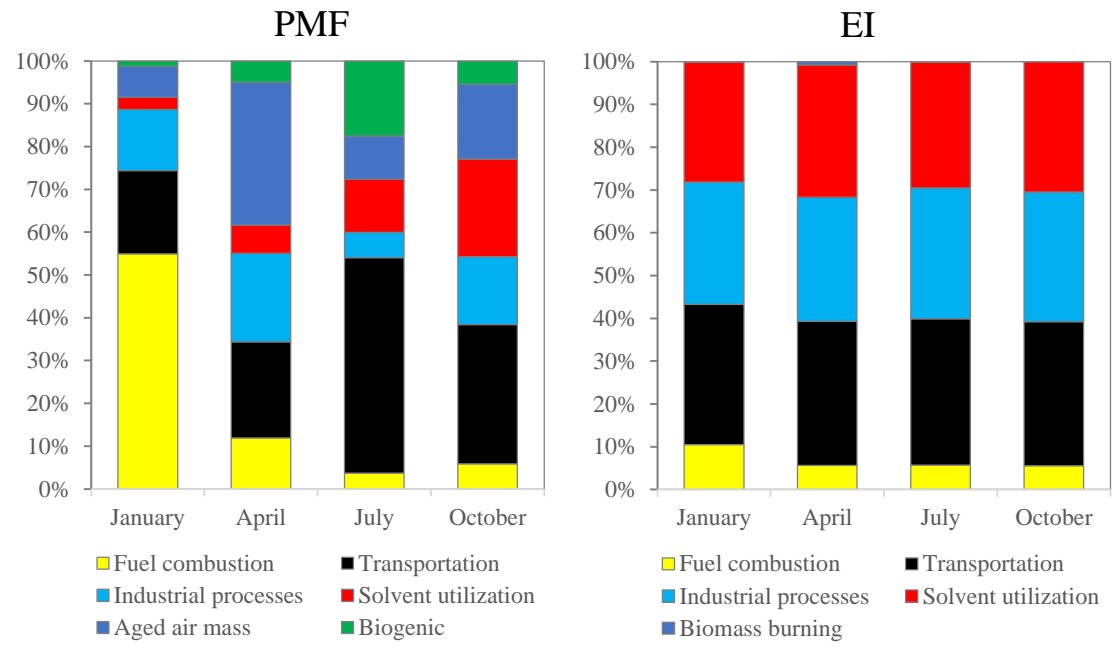

**Figure 9.Monthly VOC source structure identified by the PMF analysis (left) and the emission inventory (right).**

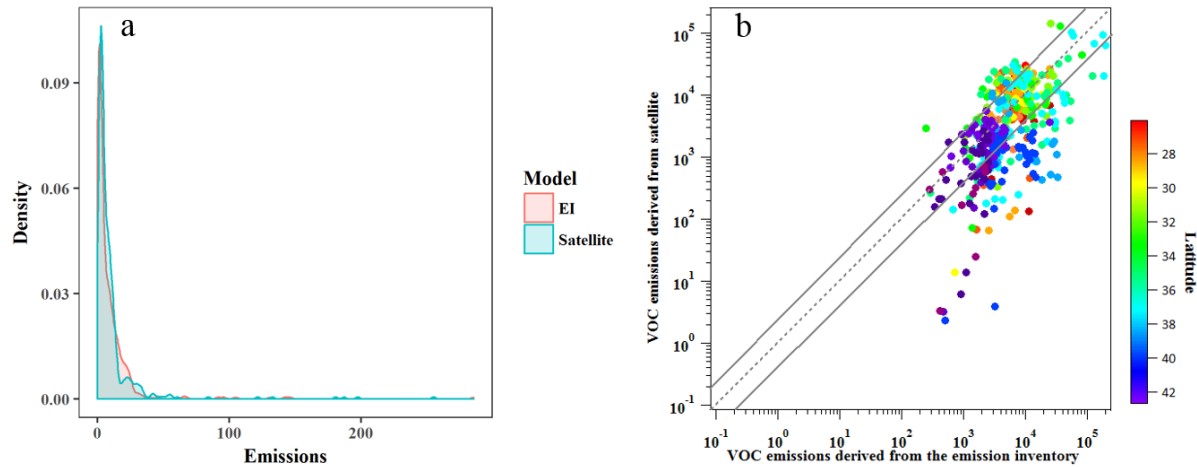

**Figure 10. Comparisons between emissions from the EF-derived emission inventory and the satellite-derived emission inventory.**

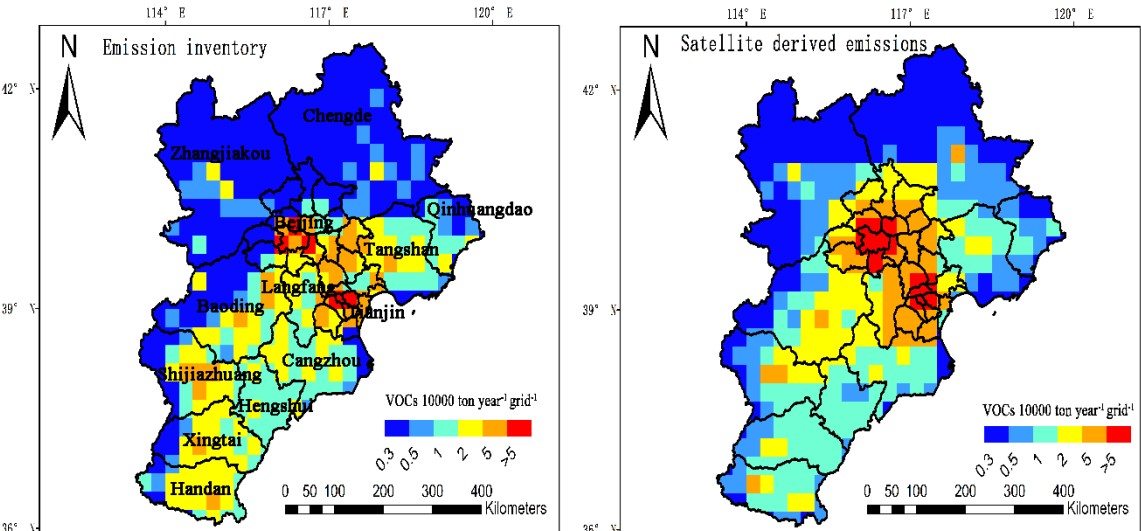

**Figure 11. Comparision between spatial distribution of VOC emissions obtained from the EF-derived emission**
**inventory and the satellite-derived emission inventory with a 0.25 $°\times$ 0.25 $°$ spatial resolution.**

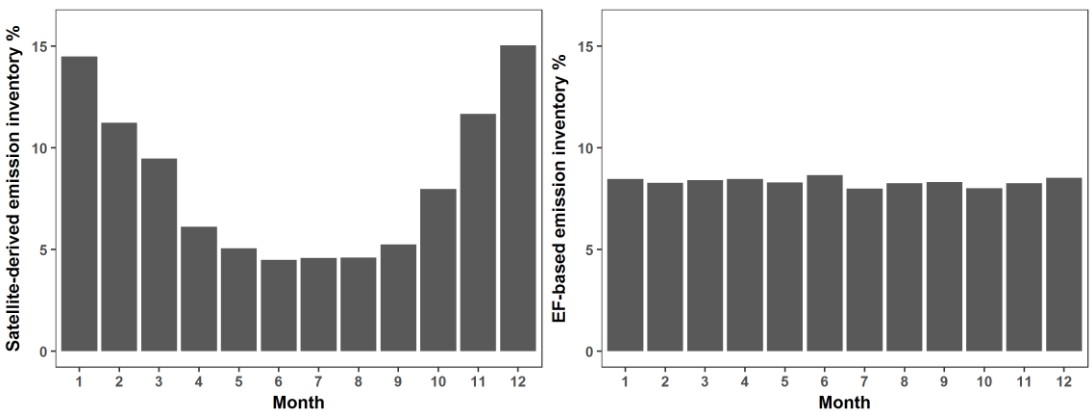

**Figure 12. Monthly variability of VOC emissions from the satellite-derived and EF-based emission inventories.**