# Peer review of "Verification of anthropogenic VOC emission inventory through ambient measurements and satellite retrievals"

_Atmospheric Chemistry and Physics, 2018_

## Referee Comment (RC1) · Anonymous Referee #1 · 27 Jan 2019

Comments to Li et al., 2019, ACPD, acp-2018-1133: Verification of anthropogenic VOC emission inventory through ambient measurements and satellite retrievals

General Description of manuscript:

The authors developed an emission inventory of anthropogenic non-methane volatile organic compounds for Beijing-Tianjin-Hebei (BTH) region of China for 2015 using emission factor approach. Their estimate of total anthropogenic VOCs emissions over BTH in 2015 is 3277.66 Gg. The authors reported that their emission inventory shows significant consistence with both ambient measurements and satellite-derived emission inventory. PMF analysis of online measurements and their emission inventory

show that vehicle emissions dominate the anthropogenic VOCs in Beijing. This study is interesting, and within the general scope of ACP. However, there are some weaknesses in current version. For example, many key statements come without citation; some references are inappropriate; some conclusions are not fully supported by their figures and numbers; some discussions are not quantitative. Therefor, I think this manuscript needs a major revision before it become suitable for publication.

General Comments:

Activity data is quite important in EF-based emission inventory. Where did the activity data come from? Can you please provide a table of activity data for each source and for each category?

What is the monthly variability of the EF-based emission inventory? And what's the difference between your monthly emissions and the satellite-derived monthly emissions?

Can you please add comparison between source structure from PMF analysis and that from your emission inventory for each season? As there might be a seasonal variability in source structure of your emission inventory.

Specific Comments:

1. Line 33: "Their direct emission sources include biogenic and anthropogenic sources". Forest fire emissions are also worth to mention here. In addition, please include some appropriate citations here.

2. Line 42-46: Zhang et al. (2009) and Li et al. (2014) have done a lot work in compiling anthropogenic VOCs emissions over China. They are also worth to cite here.

3. Line 63-66: Both Karplus et al. (2018) and Henne et al. (2016) are not appropriate references here, because Karplus et al. (2018) talks about SO2 and Henne et al. (2016) talks about methane. Please include some NMVOC-related references here.

4. Line 67-69: Please include some references here to denote "Earlier studies" and

**ACPD**
"most studies".

5. Line 105: what is "COPERT 4" short for? Please give a full name of this software when you first mention it.

6. Line 122-123: Where did "county-level", "city-level", and "provincial-level" data come from? Please include corresponding references here.

7. Line 132: Please give the source profile of Wu and Xie (2017), you can put it in the supplementary information.

8. Line 137: Please also give the height of roof site.

9. Line 169-170: Please give the emission ratios of VOC species relative to CO you obtained from the linear fit model.

10. Line 171-172: Please include references to support "(1) CO has similar sources as that of anthropogenic VOC and (2) CO emissions show lower uncertainty compared with VOC emissions".

11. Line 177: Please include appropriate references on validation of CO emission inventory of MarcoPolo Project.

12. Line 186-187: Can you please give a brief description (or formulas) on how your sampled VOCs uncertainties were calculated?

13. Line 198: "...limited..." should be "...constrained..."

14. Line 199-200: "HCHO is a high-yield product of VOCs species oxidation" should be "HCHO is a high-yield product of many VOCs species oxidation". Also, please include some appropriate references to support this sentence. Such as Millet et al. (2006), Stavrakou et al. (2015).

15. Line 200: "...relative to..." should be "...against..."

16. Line 235: "...in January, with an average value of 62.26 ppbv"? But figure 5 shows
that the January average was less than 60 ppbv.

17. Line 237: "In October, the average mixing ratio of VOCs was 50.64 ppbv". Again, the October average in figure 5 seems to be less than 50 ppbv.

18. Line 246: "The VOCs accumulated in early October decreased sharply in the middle of the month, then began to accumulate again with the change of the diffusion condition". Can you please provide some meteorological analysis or some references to support this sentence?

19. Line 265: "...a few different emission sources from CO" should be "...a few emission sources different from CO sources"

20. Line 275-277: "After the comparison with results obtained from measurements, the emissions for a majority of the non-methane hydrocarbon (NMHC) species were agreed within  $\pm 100\%$  in the emission inventory". Can you please quantify the "a majority"?

21. Line 277: "The emissions for acetonitrile came from the two methods were similar" doesn't seem to be supported by the acetonitrile emissions values (0.21 Ton yr-1 vs 16.52 Ton yr-1) listed in Table S2.

22. Line 282-283: "The annual emissions for alkanes were in agreement between the two methods". Can you please quantify the "agreement"? As the detailed emission values of many alkanes in Table S2 have large difference between these two methods.

23. Line 287: "The annual emissions for the alkenes, except ethene, correlated well". Again, please quantitatively state the "correlate well".

24. Line 288-289: "Ethene and acetylene are mainly emitted through an incomplete combustion process". Please include appropriate references for this sentence.

25. Line 293-297: What's the local VOC emission standards? Can you please give a reference?
26. Line 303: "...reaction..." should be "reactivity".

27. Line 316: "appointment" should be "apportionment"?

28. Line 323-335: Can you please add some comparison between your PMF analysis with PMF analysis from other studies during these seasons?

29. Line 384: Can you please make markers of Xingtai and Handan on the maps of figure 13?

30. Line 386-391: The subtitle of this part is "Verification of spatial distribution", but here you are discussing the monthly variabilities of satellite-derived emissions and another bottom-up emission inventory (Li et al., 2017) at length. It seems a little odd. If you want to discuss monthly variation, can you please provide the comparison between your monthly VOCs emissions and satellite-derived monthly emissions? And also, please keep the subtitle consistent with your text.

31. Line 392-393: Again, Geng et al. (2017) talks about NO2 and NOx emissions. Can you please include some appropriate references on top-down VOCs emissions and OVOC satellite observations? Such as Palmer et al.(2006), Fu et al. (2007), Marais et al.(2014), Stavrakou et al. (2015), Bauwens et al. (2016).

32. Line 407-408: Again, can you please quantify the consistence between the NMHCs emissions derived from online measurements and those from your emission inventory?

33. Table 1.: What are "1,1,2,2-" and "1,2-" in the last 3 rows?

34. Figure 2.: The unit "Ton/grid" should be "Ton year-1 grid-1".

35. Figure 4.: What's the difference between "Other VOCs" and "Others" in the bar plot? What's "Other thenes"? Why is "Others" the largest contributor in bar plot, while it is the smallest one in the pie plot?

36. Line 580-586: "...Sci Total Environ ..." in line 582, and "...Journal Of Geophysical Research-Atmospheres..." in line 585. The formats of references are not unified.
Please unify the reference format throughout the whole references.

References:

Bauwens, M., Stavrakou, T., Müller, J.-F., De Smedt, I., Van Roozendael, M., van der Werf, G. R., Wiedinmyer, C., Kaiser, J. W., Sindelarova, K., and Guenther, A.: Nine years of global hydrocarbon emissions based on source inversion of OMI formaldehyde observations, Atmos. Chem. Phys., 16, 10133-10158, https://doi.org/10.5194/acp-16-10133-2016, 2016.

Fu, T.-M., Jacob, D. J., Palmer, P. I., Chance, K., Wang, Y. X., Barletta, B., Blake, D. R., Stanton, J. C., and Pilling, M. J.: Space-based formaldehyde measurements as constraints on volatile organic compound emissions in east and south Asia and implications for ozone, J. Geophys. Res., 112, D06312, https://doi.org/10.1029/2006jd007853, 2007.

Li, M., Zhang, Q., Streets, D. G., He, K. B., Cheng, Y. F., Emmons, L. K., Huo, H., Kang, S. C., Lu, Z., Shao, M., Su, H., Yu, X., and Zhang, Y.: Mapping Asian anthropogenic emis- sions of non-methane volatile organic compounds to multiple chemical mechanisms, Atmos. Chem. Phys., 14, 5617–5638, https://doi.org/10.5194/acp-14-5617-2014, 2014.

Marais, E. A., Jacob, D. J., Guenther, A., Chance, K., Kurosu, T. P., Murphy, J. G., Reeves, C. E., and Pye, H. O. T.: Improved model of isoprene emissions in Africa using Ozone Monitoring Instrument (OMI) satellite observations of formaldehyde: implications for oxidants and particulate matter, Atmos. Chem. Phys., 14, 7693-7703, https://doi.org/10.5194/acp-14-7693-2014, 2014

Millet, D. B., Jacob, D. J., Turquety, S., Hudman, R. C., Wu, S., Fried, A., Walega, J., Heikes, B. G., Blake, D. R., Singh, H. B., Anderson, B. E., and Clarke, A. D.: Formaldehyde distribu- tion over North America: Implications for satellite retrievals of formaldehyde columns and isoprene emission, J. Geophys. Res., 111, D24S02,
https://doi.org/10.1029/2005jd006853, 2006.

Palmer, P. I., Abbot, D. S., Fu, T.-M., Jacob, D. J., Chance, K., Kurosu, T. P., Guenther, A., Wiedinmyer, C., Stanton, J. C., Pilling, M. J., Pressley, S. N., Lamb, B., and Sumner, A. L.: Quantifying the seasonal and interannual variability of North American isoprene emissions using satellite observations of the formaldehyde column, J. Geophys. Res., 111, D12315, https://doi.org/10.1029/2005jd006689, 2006.

Stavrakou, T., Müller, J.-F., Bauwens, M., De Smedt, I., Van Roozendael, M., De Mazière, M., Vigouroux, C., Hendrick, F., George, M., Clerbaux, C., Coheur, P.-F., and Guenther, A.: How consistent are top-down hydrocarbon emissions based on formaldehyde observations from GOME-2 and OMI?, Atmos. Chem. Phys., 15, 11861–11884, https://doi.org/10.5194/acp-15- 11861-2015, 2015.

Zhang, Q., Streets, D. G., Carmichael, G. R., He, K. B., Huo, H., Kannari, A., Klimont, Z., Park, I. S., Reddy, S., Fu, J. S., Chen, D., Duan, L., Lei, Y., Wang, L. T., and Yao, Z. L.: Asian emis- sions in 2006 for the NASA INTEX-B mission, Atmos. Chem. Phys., 9, 5131–5153, https://doi.org/10.5194/acp-9-5131-2009, 2009.

ACPD

---

## Referee Comment (RC2) · Anonymous Referee #3 · 2 Feb 2019

Summary and recommendation

Li et al. compile a new VOC emissions inventory for the Beijing-Tianjin-Hebei metro area, and validate it against in-situ VOC concentrations and satellite-derived emissions within the region. The new VOC observations and emission inventory are within the scope specified by ACP, representing a contribution to "substantial new data." In the present version of the manuscript there are some weaknesses in the analysis comparing the new inventory to the validation datasets. I will recommend publication once these issues are addressed.

General Comments

As the first reviewer said, the activity data is important for the construction of the inventory, and it is also not clear to me where this is sourced from. This information definitely needs to be provided in the main manuscript.

In section 3.2, much of the focus was comparing the speciation of the emission inventory based on CO ratios at the PKU site. This ultimately depends on how representative the PKU site is at the spatiotemporal scale of comparison. The PMF results remove a lot of these complications by decomposing the variability into a set of dominant modes corresponding to source types that can more easily be compared to the sector-based speciation in the inventory. This is likely a more quantitative comparison of the skill of the inventories speciation, and thus should be a greater focus.

I also so no temporal validation of the emissions inventory. The discussion of Fig. 14 in section 3.4 seems to indicate there is no temporal variation in the emissions inventory, which would be a major weakness considering the seasonal variation showed by the top-down satellite inventory. If there is no seasonality then I believe this must be included before final publication. The seasonality in total VOC emissions must be validated against the satellite inventory, and the sector based emissions can be assessed by comparing against the PMF source factor weightings.

Specific Comments

Figure 1: Please indicate the Beijing, Tianjin, and Hebei regions separately on the middle panel of the figure, as these are referenced individually throughout the text

Line 201: "Its [HCHOs] column concentration is directly related to emissions"

This also depends on the lifetime of the precursor VOC.

Line 233: "Figure 5 presents the averaging mixing ratios..."

Also indicate in the text that Figure 5 is showing observations at the PKU site

Line 241: "Figure 6 presents the time series of VOC mixing ratios . . ."

I am not sure what this paragraph/figure is getting at. There are a range of factors driving the variability in the instantaneous observations, including stability of the boundary layer, diurnal/seasonal variation in emissions, transport to the site etc. The point of using the observations is to validate the inventory. For instance, one could put the inventory into a chemical transport model and test if it can replicate the site variability in VOC concentrations. Short of doing something like this, I am not sure what the Figure is trying to show.

Line 251: "Benzene and toluene were important VOC..."

Barletta et al. (2005) discussion Benzene/Toluene ratios of different combustion sources from a survey of Chinese cities. Perhaps your discussion here can reference this in relation to the different sources.

Reference Barletta et al. (2005) "Volatile organic compounds in 43 Chinese cities" https://doi.org/10.1016/j.atmosenv.2005.06.029

Line 266: "Table 2 lists the emission ratios for individual VOCs..."

The following 2 paragraphs make the implicit assumption that emission ratios of the VOCs in the 0.25x0.25 degree grid box surround the PKU site are representative of the concentration ratios of VOCs within the site. In general I think it is difficult to assess this in a quantitative way without modelling. For instance, transport from surrounding grid boxes may be important. Also the diurnal structure of emissions will also play a role - sources that have relatively higher night-time emissions will have an outsized impact on the concentration ratio, due to the higher boundary layer stability and reduction in chemical processing. It is for reasons such as these that I find a comparison of the concentration ratios vs. emission ratios difficult.

Line 277: "the annual emissions of many OVOCs and halocarbons were much lower..."

Here it is suggested that OVOCs are underestimated by the emission inventory. However secondary production through the oxidation of precursor VOCs will have a similar

impact.

Line 317: "The PMF receptor model..."

Here in Figure 9 I would like to see a comparison between the source profiles derived from the PMF against their attributed sources from the inventory. I believe this is a more reliable test on the inventory speciation.

Line 323: "Figure 10 illustrates source contributions percentages..."

In Figure 10 it would be useful to compare the source contribution percentages derived from the PMF to those from the inventory, to address whether the inventory can or cannot replicate the temporal variations in source categories. Instead of using the Pie charts you could make a bar graph like the one in Fig. 5, putting the results of the inventory next to the observations, or just show the absolute VOC source totals as coloured lines for the four months. Doing this, you probably dont need to make the yearly comparison (Fig 11).

Line 385: "The temporal resolution of the satellite-derived emissions inventory..."

It would be useful to compare the temporal resolution of the EF-inventory to the satellite. If it is not there then it needs to be considered.

---

## Referee Comment (RC3) · Anonymous Referee #2 · 3 Feb 2019

General description: This manuscript presents work to verify anthropogenic emission inventories for the Beijing, Tianjing, Hebei area of China using a set of ambient VOC measurements made at a site in Beijing and satellite retrievals. They developed the inventory, performed a PMF analysis of the ambient data to evaluate the source structure and then evaluated the spatial distribution with satellite derived emissions. This is potentially important work a emission inventories are crucial as inputs for air quality models and thus for driving air pollution abatement strategies. It is therefore important that the inventories are accurate and comparison with emissions derived from measurements is an important tool to ensure this. The work is within the scope of ACP, however there some weaknesses that need to be addressed before final publication.

[Figure]

General comments: The emission inventory was constructed using activity data but there is no discussion as to where this came from. More detail should be given as to the source of the activity data.

Is there any temporal variation in the inventory? Presumably the activity data is time dependant (e.g. seasonal, hour of day). It would be important for the inventory to have temporal scaling factors in order for it to be used in models.

Could the authors comment on the representativeness of the PKU site for comparison with the 3kmx3km grid square of the inventory? It is difficult to use point measurements to compare to an emission rate for a much larger area so I wonder how use this comparison is?

There are many parts of the manuscript where quite vague statements on the comparison between the inventory and the measured emissions are given. For example the paragraph starting on line 274 states that a majority of NMHC agree within +- 100% with the inventory. What do the authors mean by a majority? How many agreed within 50% or 25%? Also in the paragraph starting on line 287 they state that annual emissions for alkenes, except ethene correlate well. What does 'well' mean in this case. In general the authors need to be a bit more quantitative in their statements of the degree of correlation between the inventories and the measured emissions.

Specific comments: Line 37: Better to dsay 'production' rather than 'ambient concentrations' of secondary pollutants. Line 99: Where do the emission factors (EFs) come from, please provide a reference. Line 144: What international calibration scale is the standard used for calibrating the instrument tied to? Line 175: Please provide more information here. Is there a reference for the MarcoPolo project apart from the website? What chemical transport model is used? Section 3.2.1: Quite a lot of space is given to describing the time series of the VOC measurements here but it is not really put into context with the emissions. Maybe the section could be expanded to also describe how local meteorology and long range transport affects the concentration of the VOCs as

well as their local emissions? Line 261: It is stated that all species except ïĄć-pinene and C2F2Cl3 were related to CO. Is this also true for other biogenic species such as isoprene?

---

## Author Comment (AC1) · 1 Apr 2019

**General Description of manuscript:**

The authors developed an emission inventory of anthropogenic non-methane volatile organic compounds for Beijing-Tianjin-Hebei (BTH) region of China for 2015 using emission factor approach. Their estimate of total anthropogenic VOCs emissions over BTH in 2015 is 3277.66 Gg. The authors reported that their emission inventory shows significant consistence with both ambient measurements and satellite-derived emission inventory. PMF analysis of online measurements and their emission inventory show that vehicle emissions dominate the anthropogenic VOCs in Beijing. This study is interesting, and within the general scope of ACP. However, there are some weaknesses in current version. For example, many key statements come without citation; some references are inappropriate; some conclusions are not fully supported by their figures and numbers; some discussions are not quantitative. Therefore, I think this manuscript needs a major revision before it become suitable for publication.

**Response:** Thanks a lot for your dedicated work. We really appreciate the careful reading and the useful suggestions, which help to improve the manuscript considerably. We have fully considered the comments and made revisions to our manuscript. The response and changes are listed below. The responses are in black, and revised portions are marked blue in the letter.

**General Comments:**

**1.** Activity data is quite important in EF-based emission inventory. Where did the activity data come from? Can you please provide a table of activity data for each source and for each category?

**Response:** Accepted. Thank you for your valuable suggestion. In the revised manuscript, a table containing activity data for each source and for each category and the corresponding reference was provided in the Supplementary Information (Table S2).

**2.** What is the monthly variability of the EF-based emission inventory? And what's the difference between your monthly emissions and the satellite-derived monthly emissions?

**Response:** Accepted. Thank you for your comment. The monthly VOC emissions of EF-based emission inventories can be calculated by monthly profile for each source, which usually developed based on monthly statistics (Li et al., 2017b). According to the method of Li et al. (2017b), Wu et al. (2018), and Zhang et al. (2009), we provided monthly profiles for every source in the Table S3 of the Supplementary Information.

In the revised manuscript, we provided the monthly variability of the EF-based emission inventory and discussed the difference between our monthly emissions and the satellite-derived monthly emissions. A table of monthly profiles used in this study was provided in the Supplementary Information (Table S3). Following sentences were added in section 2.1.2 to describe the method of temporal distribution, "The monthly variability of this VOC emission inventory was calculated based on the monthly profiles (Table S3). In summary, monthly profiles for industrial emissions were developed based on the monthly output of industrial products (NBS, 2015). Power plant monthly profile was derived from monthly statistics of power generation (NBS, 2015). Monthly profiles of

residential fossil fuel combustion were estimated based on household survey results (Guo et al., 2015;Zheng et al., 2014). Monthly profile of on-road vehicle emissions was derived from Li et al. (2017b). For field crop residue burning, the monthly profile was estimated based on the MODIS fire counts in croplands (Li et al., 2016). We assumed that there was no monthly variation for the emissions from the other sources (Wu and Xie, 2018)."

The monthly variability of the EF-based emission inventory obtained by this study was shown in Figure 12, which didn't exhibit obvious seasonal variations. The EF-based VOC emission inventories developed by the other studies (Li et al., 2017b; Wu and Xie, 2018) also didn't exhibit obvious seasonal variations. The discrepancies among seasons were very tiny because of little monthly variation in emissions from industrial processes, transportation, and solvent utilization (Wu and Xie, 2018). Monthly variations of the satellite-derived VOC emissions exhibit obvious seasonal characteristics, with the maximum in winter and minimum in summer, which are consistent with the seasonal characteristics of the ambient VOC mixing ratios. Thus, the satellite-derived emission inventories may better reflect the monthly characteristics of VOC emissions.

In the revised manuscript, Figure 12 was revised and comparison between the monthly variations of the satellite derived emission inventory and EF-based emission inventory was added in section 3.4 as follows, "The temporal resolution of the satellite-derived emission inventory is one month. As shown in Fig. 12, monthly variations of VOC emissions exhibit obvious seasonal characteristics, with the maximum in winter and minimum in summer, which are consistent with the seasonal characteristics of the ambient VOC mixing ratios (Fig. 5). However, monthly profiles for the EF-based emission inventory, which developed based on monthly statistics, didn't exhibit seasonal variations. EF-based VOC emission inventories developed by the other studies (Li et al., 2017; Wu and Xie, 2018) also didn't exhibit obvious seasonal variations because of little monthly variation in emissions from transportation, industrial processes, and solvent utilization (Wu and Xie, 2018). The satellite-derived emission inventories may better reflect the monthly characteristics of VOC emissions and be used to allocate monthly emissions". In addition, the subtitle of this part was changed to "Verification of spatial and temporal distributions."

References:

Guo, J., Huang, Y., and Wei, C.: North–South debate on district heating: Evidence from a household survey, Energ. Policy, 86, 295-302, 10.1016/j.enpol.2015.07.017, 2015.

Li, J., Li, Y., Bo, Y., and Xie, S.: High-resolution historical emission inventories of crop residue burning in fields in China for the period 1990–2013, Atmos. Environ., 138, 152-161, 10.1016/j.atmosenv.2016.05.002, 2016.

Li, M., Zhang, Q., Kurokawa, J.-i., Woo, J.-H., He, K., Lu, Z., Ohara, T., Song, Y., Streets, D. G., Carmichael, G. R., Cheng, Y., Hong, C., Huo, H., Jiang, X., Kang, S., Liu, F., Su, H., and Zheng, B.: MIX: a mosaic Asian anthropogenic emission inventory under the international collaboration framework of the MICS-Asia and HTAP, Atmos. Chem. Phys., 17, 935-963, 10.5194/acp-17-935-2017, 2017b.

NBS: China Economic Statistics Express, 2015.

Wu, R., and Xie, S.: Spatial Distribution of Secondary Organic Aerosol Formation Potential in China Derived from Speciated Anthropogenic Volatile Organic Compound Emissions, Environ. Sci. Technol., 52, 8146-8156, 10.1021/acs.est.8b01269, 2018.

Zheng, J., Zhang, L., Che, W., Zheng, Z., and Yin, S.: A highly resolved temporal and spatial air pollutant emission inventory for the Pearl River Delta region, China and its uncertainty assessment, Atmos. Environ., 43, 5112-5122, 10.1016/j.atmosenv.2009.04.060, 2009. Zheng, X., Wei, C., Qin, P., Guo, J., Yu, Y., Song, F., and Chen, Z.: Characteristics of residential energy consumption in China: Findings from a household survey, Energ. Policy, 75, 126-135, 10.1016/j.enpol.2014.07.016, 2014.

**3.** Can you please add comparison between source structure from PMF analysis and that from your emission inventory for each season? As there might be a seasonal variability in source structure of your emission inventory.

**Response:** Accepted. Thank you for your suggestion. As we described in the response to your General Comment No.2, monthly profiles for EF-based emission inventories usually didn't exhibit seasonal variations.

In the revised manuscript, we added comparison between source structure from PMF analysis and that from our emission inventory for each season in Figure 9, and the following sentences were added in section 3.3.2 "Compared with the seasonal PMF results, the emissions from industrial processes, transportation, and solvent utilization of the emission inventory didn't exhibit obvious seasonal variations (Fig. 9). It is because the monthly profiles of these sources, which developed on monthly statistics, have little monthly variations (Wu and Xie, 2018). The emissions from fuel combustion of the emission inventory exhibit similar seasonal variations with the PMF results with much higher emissions in winter than the other seasons. However, the relative contribution of fuel combustion for each season in the emission inventory was significantly lower than the contribution in the PMF results, especially for winter. On the basis of the above comparisons of the VOC source structure, we inferred that: (1) the annual contributions of the vehicles, solvent utilization, and industrial processes from the emission inventory and the PMF results were similar, but the monthly profiles of these sources cannot replicate the temporal variations; and (2) the fuel combustion in the PMF analysis, especially in winter, the central heating season in Beijing."

Figure 9.Monthly VOC source structure identified by the PMF analysis (left) and the emission inventory (right).

**Specific Comments:**

**1.** Line 33: "Their direct emission sources include biogenic and anthropogenic sources". Forest fire emissions are also worth to mention here. In addition, please include some appropriate citations here. **Response: Accepted.** Thank you for your suggestion. Forest fire emissions were mentioned and some appropriate citations were added here. In the revised manuscript, this sentence was revised to "Their direct emission sources include biogenic sources, forest fires, and anthropogenic sources (Guenther et al., 2006; Kansal et al., 2009; Simpson et al., 2011).

**References:**

Guenther, A., Karl, T., Harley, P., Wiedinmyer, C., Palmer, P. I., and Geron, C.: Estimates of global terrestrial isoprene emissions using MEGAN (Model of Emissions of Gases and Aerosols from Nature), Atmos. Chem. Phys., 6, 3181-3210, 2006.

Kansal, A.: Sources and reactivity of NMHCs and VOCs in the atmosphere: a review, J. Hazard. Mater., 166, 17-26, 10.1016/j.jhazmat.2008.11.048, 2009.

Simpson, I. J., Akagi, S. K., Barletta, B., Blake, N. J., Choi, Y., Diskin, G. S., Fried, A., Fuelberg, H. E., Meinardi, S., Rowland, F. S., Vay, S. A., Weinheimer, A. J., Wennberg, P. O., Wiebring, P., Wisthaler, A., Yang, M., Yokelson, R. J., and Blake, D. R.: Boreal forest fire emissions in fresh Canadian smoke plumes: C1-C10 volatile organic compounds (VOCs), CO2, CO, NO2, NO, HCN and CH3CN, Atmos. Chem. Phys., 11, 6445-6463, 10.5194/acp-11-6445-2011, 2011.

2. Line 42-46: Zhang et al. (2009) and Li et al. (2014) have done a lot work in compiling anthropogenic VOCs emissions over China. They are also worth to cite here.Response: Accepted. These studies were cited here.

**3.** Line 63-66: Both Karplus et al. (2018) and Henne et al. (2016) are not appropriate references here, because Karplus et al. (2018) talks about SO2 and Henne et al. (2016) talks about methane. Please include some NMVOC-related references here.

**Response:** Accepted. We are sorry for the inappropriate references. Fu et al. (2007) constrained the NMVOC emissions from multiple sources over East and South Asia by analyzing the spatiotemporal variability in the observed formaldehyde columns. Cao et al. (2018) used satellite retrievals along with a chemical transport model to constrain NMVOC emissions from China.

In the revised manuscript, the sentence was revised to "Since the satellite data possess the advantage of reflecting the spatial characteristics of VOCs (Fu et al., 2007), satellite-derived anthropogenic VOC emission estimations obtained from the chemical transport model can be utilized to evaluate the spatial distribution of the EF based emission inventories (Cao et al., 2018)". References:

Cao, H., Fu, T.-M., Zhang, L., Henze, D. K., Miller, C. C., Lerot, C., Abad, G. G., De Smedt, I., Zhang, Q., van Roozendael, M., Hendrick, F., Chance, K., Li, J., Zheng, J., and Zhao, Y.: Adjoint inversion of Chinese non-methane volatile organic compound emissions using space-based observations of formaldehyde and glyoxal, Atmos. Chem. Phys., 18, 15017-15046, 10.5194/acp-18-15017-2018, 2018. Fu, T.-M., Jacob, D. J., Palmer, P. I., Chance, K., Wang, Y. X., Barletta, B., Blake, D. R., Stanton, J. C., and Pilling, M. J.: Space-based formaldehyde measurements as constraints on volatile organic compound emissions in east and south Asia and implications for ozone, J. Geophys. Res., 112, 10.1029/2006jd007853, 2007.

**4.** Line 67-69: Please include some references here to denote "Earlier studies" and "most studies". **Response:** Accepted. Gaimoz et al. (2011) evaluated the source structure of VOC emission inventories by source appointment with one month ambient measurements; Borbon et al. (2013) and Wang et al. (2014) evaluated the species-specific emissions by the emission ratios to an inert tracer; Cao et al. (2018) evaluate the spatial distribution by satellite retrievals. Gaimoz et al. (2011), Borbon et al. (2013) and Wang et al (2014) were based on the data from one, one, and two-month ambient measurements, respectively. In the revised manuscript, the references to denote "Earlier studies" and "most studies" were added as follows, "Earlier studies by various research groups applied only one of these methods to evaluate either the source structure or species-specific emissions of VOC emission inventories (Gaimoz et al., 2011; Borbon et al., 2013; Wang et al. 2014; Cao et al., 2018). Moreover, most studies have been based on the data from one or two-month ambient measurements, which cannot accurately represent the annual emissions (Gaimoz et al., 2011; Borbon et al., 2014)". References:

Borbon, A., Gilman, J. B., Kuster, W. C., Grand, N., Chevaillier, S., Colomb, A., Dolgorouky, C., Gros, V., Lopez, M., Sarda-Esteve, R., Holloway, J., Stutz, J., Petetin, H., McKeen, S., Beekmann, M., Warneke, C., Parrish, D. D., and de Gouw, J. A.: Emission ratios of anthropogenic volatile organic compounds in northern mid-latitude megacities: Observations versus emission inventories in Los Angeles and Paris, J. Geophys. Res.-Atmos., 118, 2041-2057, 10.1002/jgrd.50059, 2013.

Cao, H., Fu, T.-M., Zhang, L., Henze, D. K., Miller, C. C., Lerot, C., Abad, G. G., De Smedt, I., Zhang, Q., van Roozendael, M., Hendrick, F., Chance, K., Li, J., Zheng, J., and Zhao, Y.: Adjoint inversion of Chinese non-methane volatile organic compound emissions using space-based observations of formaldehyde and glyoxal, Atmos. Chem. Phys., 18, 15017-15046, 10.5194/acp-18-15017-2018, 2018.

Gaimoz, C., Sauvage, S., Gros, V., Herrmann, F., Williams, J., Locoge, N., Perrussel, O., Bonsang, B., d'Argouges, O., Sarda-Estève, R., and Sciare, J.: Volatile organic compounds sources in Paris in spring 2007. Part II: source apportionment using positive matrix factorisation, Environ. Chem., 8, 91, 10.1071/en10067, 2011.

Wang, M., Shao, M., Chen, W., Yuan, B., Lu, S., Zhang, Q., Zeng, L., and Wang, Q.: A temporally and spatially resolved validation of emission inventories by measurements of ambient volatile organic compounds in Beijing, China, Atmos. Chem. Phys., 14, 5871-5891, 10.5194/acp-14-5871-2014, 2014.

**5.** Line 105: what is "COPERT 4" short for? Please give a full name of this software when you first mention it.

**Response:** Accepted. "COPERT" is short for "COmputer Programme to calculate Emissions from Road Transport". In the revised manuscript, the full name of the software was added as "EFs were calculated by COmputer Programme to calculate Emissions from Road Transport version 4 (COPERT 4)".

**6.** Line 122-123: Where did "county-level", "city-level", and "provincial-level" data come from? Please include corresponding references here.

**Response:** Accepted. We are sorry for the unclear descriptions. In the revised manuscript, a table containing activity data and corresponding reference for each source and for each category was added in the supplementary information (Table S2).

**7.** Line 132: Please give the source profile of Wu and Xie (2017), you can put it in the supplementary information.

**Response:** Accepted. The source profile used in this study was added in the supplementary information (Table S4). In the revised manuscript, "The source profile database used in this study was listed in Table S4." was added.

**8.** Line 137: Please also give the height of roof site.

**Response:** Accepted. In the revised manuscript, the height of the roof site was added as follow: "The roof of the Technical Physics Building at Peking University, with a height of approximately 15 m above the ground (PKU, 39.99 N, 116.33 E, Fig. 1) was selected as the sampling site".

**9.** Line 169-170: Please give the emission ratios of VOC species relative to CO you obtained from the linear fit model.

**Response:** Accepted. The emission ratios of VOC species relative to CO obtained from the linear fit model were listed in Table S5.

**10.** Line 171-172: Please include references to support "(1) CO has similar sources as that of anthropogenic VOC and (2) CO emissions show lower uncertainty compared with VOC emissions". **Response: Accepted.** Thank you for your suggestion. A lot of studies have chosen CO as the reference compound to study the emission ratios for many cities in the world (Warneke et al., 2007; Coll et al., 2010; Borbon et al., 2013; Wang et al., 2014). Wang et al. (2014) calculated the emission ratios for Beijing and described that CO has similar sources as that of anthropogenic VOC in Beijing and the uncertainty of CO emissions are lower than that of VOC emissions in detailed. In the revised manuscript, references were added as follows "In this study, we selected CO as a reference compound considering that: (1) CO has similar sources as that of anthropogenic VOC and (2) CO emissions show lower uncertainty compared with VOC emissions (Warneke et al., 2007; Wang et al., 2014). Thus, CO was a suitable reference compound (Coll et al., 2010;Borbon et al., 2013;Wang et al., 2014)". References:

Borbon, A., Gilman, J. B., Kuster, W. C., Grand, N., Chevaillier, S., Colomb, A., Dolgorouky, C., Gros, V., Lopez, M., Sarda-Esteve, R., Holloway, J., Stutz, J., Petetin, H., McKeen, S., Beekmann, M., Warneke, C., Parrish, D. D., and de Gouw, J. A.: Emission ratios of anthropogenic volatile organic compounds in northern mid-latitude megacities: Observations versus emission inventories in Los Angeles and Paris, J. Geophys. Res.-Atmos., 118, 2041-2057, 10.1002/jgrd.50059, 2013.

Wang, M., Shao, M., Chen, W., Yuan, B., Lu, S., Zhang, Q., Zeng, L., and Wang, Q.: A temporally and spatially resolved validation of emission inventories by measurements of ambient volatile organic compounds in Beijing, China, Atmos. Chem. Phys., 14, 5871-5891, 10.5194/acp-14-5871-2014, 2014.

Coll, I., Rousseau, C., Barletta, B., Meinardi, S., and Blake, D. R.: Evaluation of an urban NMHC emission inventory by measurements and impact on CTM results, Atmos. Environ., 44, 3843-3855, 10.1016/j.atmosenv.2010.05.042, 2010.

Warneke, C., McKeen, S. A., de Gouw, J. A., Goldan, P. D., Kuster, W. C., Holloway, J. S., Williams, E. J., Lerner, B. M., Parrish, D. D., Trainer, M., Fehsenfeld, F. C., Kato, S., Atlas, E. L., Baker, A., and Blake, D. R.: Determination of urban volatile organic compound emission ratios and comparison with an emissions database, J. Geophys. Res., 112, 10.1029/2006jd007930, 2007.

**11.** Line 177: Please include appropriate references on validation of CO emission inventory of MarcoPolo Project.

**Response:** Accepted. Thank you for your suggestion. MarcoPolo project is a collaborative research project by Chinese and European partners to study the emission sources using state-of-the-art techniques and to provide the latest air up-to-data emission inventories for China. For the CO emission inventory, MarcoPolo project copied the Multi-resolution Emission Inventory for China (MEIC) emissions (Hooyberghs et al., 2016). The MEIC is a uniform emission model framework developed Tsinghua University to estimate anthropogenic emissions China by over (http://www.meicmodel.org/index.html). The CO emission inventory of MEIC has been validated by the chemical transport model (Hu et al., 2017), satellite observations (Yumimoto et al. 2014), and comparison with other studies (Li et al., 2017a).

In the revised manuscript, appropriate references was added as follows: "The annual emission value of CO was obtained from the CO emission inventory of the MarcoPolo Project (http://www.marcopolo-panda.eu, Hooyberghs et al., 2016), which was copied from the Multi-resolution Emission Inventory for China (MEIC) emissions (http://www.meicmodel.org/index.html). This emission inventory has been validated by the chemical transport model (Hu et al., 2017), satellite observations (Yumimoto et al. 2014), and comparison with other studies (Li et al., 2017a)". References:

Hooyberghs, H., Veldeman, N., and Maiheu, B.: Marco Polo Emission Inventory for East-China: Basic Description, 2016.

Hu, J., Li, X., Huang, L., Ying, Q., Zhang, Q., Zhao, B., Wang, S., and Zhang, H.: Ensemble prediction of air quality using the WRF/CMAQ model system for health effect studies in China, Atmos. Chem. Phys., 17, 13103-13118, 10.5194/acp-17-13103-2017, 2017.

Li, M., Liu, H., Geng, G., Hong, C., Liu, F., Song, Y., Tong, D., Zheng, B., Cui, H., Man, H., Zhang, Q., and He, K.: Anthropogenic emission inventories in China: a review, National Science Review, 4, 834-866, 10.1093/nsr/nwx150, 2017a.

Yumimoto, K., Uno, I., and Itahashi, S.: Long-term inverse modeling of Chinese CO emission from satellite observations, Environ. Pollut., 195, 308-318, 10.1016/j.envpol.2014.07.026, 2014.

**12.** Line 186-187: Can you please give a brief description (or formulas) on how your sampled VOCs uncertainties were calculated?

**Response:** Accepted. The observed uncertainty file was set following the method proposed by Polissar et al. (1998), which was recommended by the user guide of the PMF model. In the revised manuscript, the description was added as follows: "The observed uncertainty file was set following the method proposed by Polissar et al. (1998), which was recommended by the user guide of the PMF model. The uncertainty is calculated by Eq. (3), if the mixing ratio is equal to or less than the MDL; the uncertainty is calculated using Eq. (4), if mixing ratio if larger than the MDL (USEPA, 2014a).

Uncertainty
$$=\frac{5}{6} \times MDL$$
 Eq. (3)

Uncertainty =
$$\sqrt{(Error \ Fraction \ \times \ mixing \ ratio)^2 + (0.5 \ \times \ MDL)^2}$$
 Eq. (4)

References:

Polissar, A. V., Hopke, P. K., Paatero, P., Malm, W. C., and Sisler, J. F.: Atmospheric aerosol over Alaska: 2. Elemental composition and sources, J. Geophys. Res.-Atmos., 103, 19045-19057, 10.1029/98jd01212, 1998.

USEPA: Positive Matrix Factorization (PMF) 5.0 Fundamentals and User Guide, US Environmental Protection Agency, Office of Research and Development, Washington, D.C, 2014a.

**13.** Line 198: "...limited..." should be "...constrained..." **Response: Accepted.** "limited" was changed to "constrained".

**14.** Line 199-200: "HCHO is a high-yield product of VOCs species oxidation" should be "HCHO is a high-yield product of many VOCs species oxidation". Also, please include some appropriate references to support this sentence. Such as Millet et al. (2006), Stavrakou et al. (2015).

**Response:** Accepted. "HCHO is a high-yield product of VOCs species oxidation" was revised to "HCHO is a high-yield product of many VOCs species oxidation". And some appropriate references such as Millet et al. (2006), Stavrakou et al. (2015) were added.

**15.** Line 200: ". . . relative to. . ." should be ". . . against. . ." **Response: Accepted.** "relative to" was changed to "against".

**16.** Line 235: ". . . in January, with an average value of 62.26 ppbv"? But figure 5 shows that the January average was less than 60 ppbv.

**Response:** Accepted. Thank you for your comment. Figure 5 does not show the distribution of the data well. The following box-plot shows the distributions of the VOC mixing ratios in January, April, July, and October 2015. As shown in this figure, the January average was larger than 60 ppbv (red dot). In the revised manuscript, the box-plot was added in the Supplementary Information as Fig. S4.

---

## Author Comment (AC2) · 1 Apr 2019

**Summary and recommendation**

Li et al. compile a new VOC emissions inventory for the Beijing-Tianjin-Hebei metro area, and validate it against in-situ VOC concentrations and satellite-derived emissions within the region. The new VOC observations and emission inventory are within the scope specified by ACP, representing a contribution to "substantial new data." In the present version of the manuscript there are some weaknesses in the analysis comparing the new inventory to the validation datasets. I will recommend publication once these issues are addressed.

**Response:** Thanks a lot for your dedicated work. We really appreciate the careful reading and the useful suggestions which help to improve the manuscript considerably. We have fully considered the comments and made revisions to our manuscript. The response and changes are listed below. The responses are in black, and revised portions are marked blue in the letter.

**General Comments**

**1.** As the first reviewer said, the activity data is important for the construction of the inventory, and it is also not clear to me where this is sourced from. This information definitely needs to be provided in the main manuscript.

**Response: Accepted.** Thank you for your valuable suggestion. In the revised a table containing activity data for each source and for each category and the corresponding reference was provided in the Supplementary Information (Table S2).

**2.** In section 3.2, much of the focus was comparing the speciation of the emission inventory based on CO ratios at the PKU site. This ultimately depends on how representative the PKU site is at the spatiotemporal scale of comparison. The PMF results remove a lot of these complications by decomposing the variability into a set of dominant modes corresponding to source types that can more easily be compared to the sector-based speciation in the inventory. This is likely a more quantitative comparison of the skill of the inventories speciation, and thus should be a greater focus.

**Response: Accepted.** Thank you for your valuable suggestion. The approach for calculation of VOC emissions based on ambient measurements has several limitations and uncertainties. The description of the method was unclear. In the revised manuscript, we revised the description of the method and added some discussion about the evaluation method in section 2.3.1.Please refer to the response to **specific comment #6** for detailed changes. We agreed that PMF results are a reliable test on the inventory speciation. According to your suggestion, we also compared the source profiles derived from the PMF against their attributed sources from the inventory to test on the inventory speciation. Please refer to the response to **specific comment # 8** for detailed changes.

**3.** I also so no temporal validation of the emissions inventory. The discussion of Fig. 14 in section 3.4 seems to indicate there is no temporal variation in the emissions inventory, which would be a major weakness considering the seasonal variation showed by the top-down satellite inventory. If there is no seasonality then I believe this must be included before final publication. The seasonality in total VOC emissions must be validated against the satellite inventory, and the sector based emissions can be

assessed by comparing against the PMF source factor weightings.

**Response: Accepted.** Thank you for your valuable suggestion. Because EF-based VOC emission inventories usually didn't exhibit obvious seasonal variations for little monthly variation in monthly statistics (Li et al., 2017; Wu and Xie, 2018). Therefore, we discussed the monthly variabilities of satellite-derived emissions and another bottom-up emission inventory (Li et al., 2017). In the revised manuscript, we provided the monthly variability of our EF-based emission inventory and it was validated against the satellite inventory. We also validated the seasonal source structure of our EF-based emissions against the PMF source factor results.

The monthly profile can calculate the monthly VOC emissions of EF-based emission inventories for each source, which usually developed based on monthly statistics (Li et al., 2017b). According to the method of Li et al. (2017b), Wu et al. (2018), and Zhang et al. (2009), we provided monthly profiles for every source in the Table S3 of the Supplementary Information. In the revised manuscript, a table of monthly profiles used in this study was provided in the Supplementary Information (Table S3). Following sentences were added in section 2.1.2 to describe the method of temporal distribution, "The monthly variability of this VOC emission inventory was calculated based on the monthly profiles (Table S3). In summary, monthly profiles for industrial emissions were developed based on the monthly output of industrial products (NBS, 2015). Power plant monthly profile was derived from monthly statistics of power generation (NBS, 2015). Monthly profiles of residential fossil fuel combustion were estimated based on household survey results (Guo et al., 2015; Zheng et al., 2014). Monthly profile of on-road vehicle emissions was derived from Li et al. (2017b). For field crop residue burning, the monthly profile was estimated based on the MODIS fire counts in croplands (Li et al., 2016). We assumed that there was no monthly variation for the emissions from the other sources (Wu and Xie, 2018)."

The monthly variability of the EF-based emission inventory obtained by this study was shown in Figure 12, which didn't exhibit apparent seasonal variations. The EF-based VOC emission inventories developed by the other studies (Li et al., 2017b; Wu and Xie, 2018) also didn't exhibit obvious seasonal variations. The discrepancies among seasons were very tiny because of little monthly variation in emissions from industrial processes, transportation, and solvent utilization (Wu and Xie, 2018). Monthly variations of the satellite-derived VOC emissions exhibit obvious seasonal characteristics, with the maximum in winter and minimum in summer, which are consistent with the seasonal characteristics of the ambient VOC mixing ratios. Thus, the satellite-derived emission inventories may better reflect the monthly characteristics of VOC emissions.

In the revised manuscript, Figure 12 was revised, and comparison between the monthly variations of the satellite-derived emission inventory and EF-based emission inventory was added in section 3.4 as follows "The temporal resolution of the satellite-derived emission inventory is one month. As shown in Fig. 12, monthly variations of VOC emissions exhibit obvious seasonal characteristics, with the maximum in winter and minimum in summer, which are consistent with the seasonal characteristics of the ambient VOC mixing ratios (Fig. 5). However, monthly profiles for the EF-based emission inventory, which developed based on monthly statistics, didn't exhibit seasonal variations. EF-based VOC emission inventories developed by the other studies (Li et al., 2017; Wu and Xie, 2018) also didn't exhibit obvious seasonal variations because of little monthly variation in emissions from transportation, industrial processes, and solvent utilization (Wu and Xie, 2018). The satellite-derived

emission inventories may better reflect the monthly characteristics of VOC emissions and be used to allocate monthly emissions". In addition, the subtitle of this part was changed to "Verification of spatial and temporal distributions".

[Figure]

**Figure 12. Monthly variability of VOC emissions from the satellite-derived and EF-based emission inventories.**

In the revised manuscript, we added comparison between source structure from PMF analysis and that from our emission inventory for each season in Figure 9. The following sentences were added in section 3.3.2, "Compared with the seasonal PMF results, the emissions from industrial processes, transportation, and solvent utilization of the emission inventory didn't exhibit obvious seasonal variations (Fig. 9). It is because the monthly profiles of these sources, which developed on monthly statistics, have little monthly variations (Wu and Xie, 2018). The emissions from fuel combustion of the emission inventory exhibit similar seasonal variations with the PMF results with much higher emissions in winter than the other seasons. However, the relative contribution of fuel combustion for each season in the emission inventory was significantly lower than the contribution in the PMF results, especially for winter. On the basis of the above comparisons of the VOC source structure, we inferred that: (1) the annual contributions of the vehicles, solvent utilization, and industrial processes from the emission inventory and the PMF results were similar, but the monthly profiles of these sources cannot replicate the temporal variations; and (2) the fuel combustion in the emission inventory showed a considerably lower relative contribution than the value from the PMF analysis, especially in winter, the central heating season in Beijing."

[Figure]

**Figure 9.Monthly VOC source structure identified by the PMF analysis (left) and the emission inventory (right).**

**Response: Accepted.** Thank you for your comment. We are sorry for the unclear description of the method we used. We agreed that the VOC concentrations obtained from field observations could not be directly compared with the VOC emissions of the grid box due to some physical (pollutant transport, change of boundary layer height or sedimentation) and chemical transformation processes. There are some commonly used methods can help to compare the VOC concentrations and emissions, including chemical transport model simulation, satellite inversion, receptor modelling, and the "emission ratio" method (Borbon et al., 2013).

In this study, we used the "emission ratio" method to verify the VOC emissions. This method has been widely used to evaluate emission inventories of VOCs (Fu et al., 2007; Hsu et al., 2010; Shao et al., 2011; Borbon et al., 2013; Wang et al., 2014). The theory of this method is that the relative ratios between enhancements of VOCs and the increasing of a trace gas (such as CO) could reflect the ratios of their emission strength (Shao et al., 2011). The relative ratios can help reduce the influence of physical transformation processes. Thus, if the emissions of the trace gas can be determined, we can calculate out the emissions of VOC species (Eq. 2 of our manuscript). In order to reduce the influence of the chemical process, the ratios should be calculated excluding or correcting the effect of photochemical processing on measured VOC (Wang et al., 2014). A commonly used method to determine emission ratios is to utilize a linear regression fit, and temporal filters (nighttime) can be applied to account for the influence of chemistry (Borbon et al., 2013).

In this study, we choose CO as the trace gas, and we use the nighttime linear regression fit method to determine the emission ratios, then we calculated the emissions of individual VOC species based on the emission ratios and CO emissions. According to Wang et al. (2014), the local time period 03:00 to 07:00 was set as a temporal filter to reduce the impact of photochemical processing. The PKU site has been used to represent a typical urban environment in Beijing in many studies (Song et al., 2007; Yuan et al., 2012; Wang et al., 2014; Li et al., 2015; Wu et al., 2016). Wang et al. (2014) measured the VOC mixing ratio during summer and winter in the PKU site, calculated the emissions of individual

VOC species, and compared the emissions with the TRACE-P (Streets et al., 2003) and INTEX-B (Zhang et al., 2009) emission inventories. Since the online VOC measurement is difficult and costly, lots studies only use the VOC mixing ratios of one online sampling site to evaluate the emissions of a larger area. Borbon et al. (2013) compared the emission results of the "emission ratio" method with emission inventories of Los Angeles based on observations in one site. Shao et al. (2011) also use one site to represent the concentrations of one city. Although the online VOC measurement is difficult and costly; we recommend more online VOC measurements be conducted.

It should be noted that this approach for the calculation of VOC emissions based on ambient measurements has several limitations and uncertainties. First, in this study, we evaluated the emission inventory based on VOC measurement at one site, which limits the spatial representation of VOC measurement data relative to those observations in more sites. Secondly, we assume that the air mass over the site could respect the average emissions of the grid box, which will lead some uncertainties. In addition, this approach relies on the assumption that the composition of urban emissions relative to CO. Chemical transport model simulation is an ideal approach to verify emission inventories based on field observations. We will recommend and try to use this approach to verify the emissions in future study. According to your suggestion, we also compared the source profiles derived from the PMF against their attributed sources from the inventory to test on the inventory speciation. Please refer to the response to specific comment # 8.

In order to make it clear, in the revised manuscript, more detailed description of the method and the limitation was added in section 2.3.1 as follows:

[revised manuscript text omitted]

Zhang, Q., Yuan, B., Shao, M., Wang, X., Lu, S., Lu, K., Wang, M., Chen, L., Chang, C. C., and Liu, S. C.: Variations of ground-level $O_3$ and its precursors in Beijing in summertime between 2005 and 2011, Atmos. Chem. Phys., 14, 6089-6101, 10.5194/acp-14-6089-2014, 2014.

**7.** Line 277: "the annual emissions of many OVOCs and halocarbons were much lower..." Here it is suggested that OVOCs are underestimated by the emission inventory. However secondary production through the oxidation of precursor VOCs will have a similar impact.

**Response: Accepted.** Thank you for your comment. As described in section 2.3.1, in order to reduce the impact of secondary production, the local time period 03:00 to 07:00 was set as a temporal filter. The emission ratios of VOC species to the reference compound (ERVOC) from 03:00 to 07:00 local time were estimated using the linear fit model.

While we try to reduce the impact of secondary production, but it still will have some impact. In the revised manuscript, following sentence was added in section 3.2.2 "In addition, secondary production through the oxidation of precursor VOCs might impact the accuracy of OVOC emission estimated from the observations".

**8.** Line 317: "The PMF receptor model. . ." Here in Figure 9 I would like to see a comparison between the source profiles derived from the PMF against their attributed sources from the inventory. I believe this is a more reliable test on the inventory speciation.

**Response: Accepted.** Thank you for your valuable suggestion. In the revised manuscript, Figure 9 was removed to the Supplementary Information as Figure S5. Comparison of source profiles for VOCs calculated by PMF and the emission inventory was presented in Figure 8. The following sentences were added "Figure 8 illustrates the comparison between source profiles derived from the PMF against their attributed sources from the emission inventory. The source profiles for fuel combustion agreed between the two methods. For the source profiles of transportation, the contributions of C2 – C4 alkanes derived from the PMF were larger than contributions from the emission inventory. Aromatics were the dominant group in the source profiles for solvent utilization derived from the two methods. For the source profiles of industrial processes, the proportions of some halocarbons OVOCs from the PMF were larger than proportions from the emission inventory. The finding agreed with the results of section 3.2.2 that the proportions of OVOCs and halocarbons in the source profile database may be

unreliable."

[Figure]

**Figure 8. Source profiles for VOCs calculated by PMF and the emission inventory (Species name was shown in Table S4) .**

**9.** Line 323: "Figure 10 illustrates source contributions percentages..." In Figure 10 it would be useful to compare the source contribution percentages derived from the PMF to those from the inventory, to address whether the inventory can or cannot replicate the temporal variations in source categories. Instead of using the Pie charts you could make a bar graph like the one in Fig. 5, putting the results of the inventory next to the observations, or just show the absolute VOC source totals as coloured lines for the four months. Doing this, you probably don't need to make the yearly comparison (Fig 11).

**Response: Accepted.** Thank you for your valuable suggestion. In the revised manuscript, Figure 10 was revised accordingly. Figure 11 was moved to the Supplementary Information, and the following sentences were added in section 3.3.2 "Compared with the seasonal PMF results, the emissions from industrial processes, transportation, and solvent utilization of the emission inventory didn't exhibit obvious seasonal variations (Fig. 9). It is because the monthly profiles of these sources, which developed on monthly statistics, have little monthly variations in the monthly profiles (Wu and Xie, 2018). The emissions from fuel combustion of the emission inventory exhibit similar seasonal

variations with the PMF results with much higher emissions in winter than the other seasons. However, the relative contribution of fuel combustion for each season in the emission inventory was significantly lower than the contribution in the PMF results, especially for winter. On the basis of the above comparisons of the VOC source structure, we inferred that: (1) the annual contributions of the vehicles, solvent utilization, and industrial processes from the emission inventory and the PMF results were similar, but the monthly profiles of these sources cannot replicate the temporal variations; and (2) the fuel combustion in the emission inventory showed a considerably lower relative contribution than the value from the PMF analysis, especially in winter, the central heating season in Beijing."

[Figure]

**Figure 9.Monthly VOC source structure identified by the PMF analysis (left) and the emission inventory (right).**

**10.** Line 385: "The temporal resolution of the satellite-derived emissions inventory. . ." It would be useful to compare the temporal resolution of the EF-inventory to the satellite. If it is not there then it needs to be considered.

**Response: Accepted.** Thank you for your valuable suggestion. Because EF-based VOC emission inventories usually didn't exhibit obvious seasonal variations because of little monthly variation in monthly statistics (Li et al., 2017; Wu and Xie, 2018). Therefore, we discussed the monthly variabilities of satellite-derived emissions and another bottom-up emission inventory (Li et al., 2017). In the revised manuscript, we provided the monthly variability of our EF-based emission inventory, and it was validated against the satellite inventory.

The monthly VOC emissions of EF-based emission inventories can be calculated by the monthly profile for each source, which usually developed based on monthly statistics (Li et al., 2017b). According to the method of Li et al. (2017b), Wu et al. (2018), and Zhang et al. (2009), we provided monthly profiles for every source in the Table S3 of the Supplementary Information. In the revised manuscript, a table of monthly profiles used in this study was provided in the Supplementary Information (Table S3). Following sentences were added in section 2.1.2 to describe the method of temporal distribution, "The monthly variability of this VOC emission inventory was calculated based

on the monthly profiles (Table S3). In summary, monthly profiles for industrial emissions were developed based on the monthly output of industrial products (NBS, 2015). Power plant monthly profile was derived from monthly statistics of power generation (NBS, 2015). Monthly profiles of residential fossil fuel combustion were estimated based on household survey results (Guo et al., 2015;Zheng et al., 2014). Monthly profile of on-road vehicle emissions was derived from Li et al. (2017b). For field crop residue burning, the monthly profile was estimated based on the MODIS fire counts in croplands (Li et al., 2016). We assumed that there was no monthly variation for the emissions from the other sources (Wu and Xie, 2018)."

The monthly variability of the EF-based emission inventory obtained by this study was shown in Figure 12, which didn't exhibit obvious seasonal variations. The EF-based VOC emission inventories developed by the other studies (Li et al., 2017b; Wu and Xie, 2018) also didn't exhibit obvious seasonal variations. The discrepancies among seasons were very tiny because of little monthly variation in emissions from industrial processes, transportation, and solvent utilization (Wu and Xie, 2018). Monthly variations of the satellite-derived VOC emissions exhibit obvious seasonal characteristics, with the maximum in winter and minimum in summer, which are consistent with the seasonal characteristics of the ambient VOC mixing ratios. Thus, the satellite-derived emission inventories may better reflect the monthly characteristics of VOC emissions.

In the revised manuscript, Figure 12 was revised and comparison between the monthly variations of the satellite derived emission inventory and EF-based emission inventory was added in section 3.4 as follows, "The temporal resolution of the satellite-derived emission inventory is one month. As shown in Fig. 12, monthly variations of VOC emissions exhibit obvious seasonal characteristics, with maximum the in winter and minimum in summer, which are consistent with the seasonal characteristics of the ambient VOC mixing ratios (Fig. 5). However, monthly profiles for the EF-based emission inventory, which developed based on monthly statistics, didn't exhibit seasonal variations. EF-based VOC emission inventories developed by the other studies (Li et al., 2017; Wu and Xie, 2018) also didn't exhibit obvious seasonal variations because of little monthly variation in emissions from transportation, industrial processes, and solvent utilization (Wu and Xie, 2018). The satellite-derived emission inventories may better reflect the monthly characteristics of VOC emissions and be used to allocate monthly emissions". In addition, the subtitle of this part was changed to "Verification of spatial and temporal distributions."

[Figure]

**Figure 12. Monthly variability of VOC emissions from the satellite-derived and EF-based emission inventories.**

---

## Author Comment (AC3) · 1 Apr 2019

**General description:**

This manuscript presents work to verify anthropogenic emission inventories for the Beijing, Tianjing, Hebei area of China using a set of ambient VOC measurements made at a site in Beijing and satellite retrievals. They developed the inventory, performed a PMF analysis of the ambient data to evaluate the source structure and then evaluated the spatial distribution with satellite derived emissions. This is potentially important work a emission inventories are crucial as inputs for air quality models and thus for driving air pollution abatement strategies. It is therefore important that the inventories are accurate and comparison with emissions derived from measurements is an important tool to ensure this. The work is within the scope of ACP, however there some weaknesses that need to be addressed before final publication.

**Response:** Thanks a lot for your dedicated work. We really appreciate the careful reading and the useful suggestions, which help to improve the manuscript considerably. We have fully considered the comments and made revisions to our manuscript. The response and changes are listed below. The responses are in black, and revised portions are marked blue in the letter.

**General comments:**

**1.** The emission inventory was constructed using activity data but there is no discussion as to where this came from. More detail should be given as to the source of the activity data.

**Response:** Accepted. Thank you for your valuable suggestion. In the revised a table containing activity data for each source and for each category and the corresponding reference was provided in the Supplementary Information (Table S2).

**2.** Is there any temporal variation in the inventory? Presumably the activity data is time dependant (e.g. seasonal, hour of day). It would be important for the inventory to have temporal scaling factors in order for it to be used in models.

**Response:** Accepted. Thank you for your valuable suggestion. The monthly activity data is available. In the revised manuscript, we provided the monthly variability of the EF-based emission inventory.

The monthly VOC emissions of EF-based emission inventories can be calculated by monthly profile for each source, which usually developed based on monthly statistics (Li et al., 2017b). According to the method of Li et al. (2017b), Wu et al. (2018), and Zhang et al. (2009), we provided monthly profiles for every sources in the Table S3 of the Supplementary Information. In the revised manuscript, a table of monthly profiles used in this study was provided in the Supplementary Information (Table S3). Following sentences were added in section 2.1.2 to describe the method of temporal distribution, "The monthly variability of this VOC emission inventory was calculated based on the monthly profiles (Table S3). In summary, monthly profiles for industrial emissions were developed based on monthly output of industrial products (NBS, 2015). Power plant monthly profile was derived from monthly statistics of power generation (NBS, 2015). Monthly profiles of residential fossil fuel combustion were estimated based on household survey results (Guo et al., 2015;Zheng et al., 2014). Monthly profile of on-road vehicle emissions was derived from Li et al. (2017b). For field crop residue burning, the monthly profile was estimated based on the MODIS fire counts in croplands (Li et al., 2016). We assumed that there was no monthly variation for the emissions from the other sources (Wu and Xie, 2018)." The monthly variability of the EF-based emission inventory obtained by this study was added in Figure 12, which didn't exhibit obvious seasonal variations. The EF-based VOC emission inventories developed by the other studies (Li et al., 2017b; Wu and Xie, 2018) also didn't exhibit obvious seasonal variations. The discrepancies among seasons were very small because of little monthly variation in emissions from industrial processes, transportation, and solvent utilization (Wu and Xie, 2018).

**Response:** Accepted. Thank you for your valuable suggestion. As we descripted in Section 3.2.1 "The VOCs accumulated in early October decreased sharply in the middle of the month, then began to accumulate again with the change of the diffusion condition". Local meteorology could impact the VOCs mixing ratios. Time series of wind speed and VOC mixing ratios in October 2015 can support the hypothesis. However, according to the comment of the other review, the point of using the observations is to validate the inventory and this section should be concise. In order to focus on the topic of this study, we expanded this section in the Supplementary Information.

Combined with reviewers' comments, in the revised manuscript, the second paragraph of Section 3.2.1 and Figure 6 was moved to the Supplementary Information. The impact of local meteorology on the mixing ratio of VOCs was added in the paragraph, which was revised to "Figure S2 presents the time series of VOC mixing ratios. The mixing ratios of VOCs in January were variable, with maximum value of 245.54 ppbv. There were lots of periods with high VOC mixing ratios in January. In April, the average VOC mixing ratio was not as high as in January but the mixing ratios of VOCs change a lot, a maximum value of 150.24 ppbv. The mixing ratios of VOCs in July were stable, with the highest level of 92.28 ppbv. The highest VOC mixing ratio in October was 201.10 ppbv. In early October, the VOCs accumulated when the wind speed was low (Fig. S3). Then VOCs decreased sharply when the wind speed became higher. And the VOC began to accumulate again with change of the wind speed. This shows that local meteorology could affect the mixing ratios of VOCs. "

Figure S3. Time series of wind speed and VOC mixing ratios in October 2015.

**6.** Line 261: It is stated that all species except  $\beta$ -pinene and C2F2Cl3 were related to CO. Is this also true for other biogenic species such as isoprene?

**Response:** Accepted. Thank you for your comment. We stated that all species except  $\beta$ -pinene and

C2F2Cl3 were significantly related to CO (p < 0.05), which based on the p-value of the Pearson correlation. A p-value of less than 0.05 indicates there is a correlation relationship between CO and this species. The p-value for isoprene is less than 0.01, which means the Pearson correlation of isoprene and CO is significant at the 0.01 level (two-tailed). Isoprene is a topic biogenic species, however, Isoprene was detected at high percentage in vehicular exhaust as well (Liu et al., 2008). So isoprene was related to CO in urban area (Wang et al., 2014). Although most species were related to CO (p < 0.05), the R-values (correlation coefficients) for each species can be vary widely. Affected by the biogenic emissions, Isoprene had a strong correlation with CO in winter (R=0.77), and had a weaker correlation with CO in summer (R=0.18).

To make it clearer, in the revised manuscript, following sentence was added in section 3.2.2 "Affected by the biogenic emissions (Guenther et al., 2006), isoprene had a stronger correlation with CO in winter (R=0.77), and had a weaker correlation with CO in summer (R=0.18)". References:

Guenther, A., Karl, T., Harley, P., Wiedinmyer, C., Palmer, P. I., and Geron, C.: Estimates of global terrestrial isoprene emissions using MEGAN (Model of Emissions of Gases and Aerosols from Nature), Atmos. Chem. Phys., 6, 3181-3210, 2006.

Liu, Y., Shao, M., Fu, L. L., Lu, S. H., Zeng, L. M., and Tang, D. G.: Source profiles of volatile organic compounds (VOCs) measured in China: Part I, Atmos. Environ., 42, 6247-6260, 10.1016/j.atmosenv.2008.01.070, 2008.

Wang, M., Shao, M., Chen, W., Yuan, B., Lu, S., Zhang, Q., Zeng, L., and Wang, Q.: A temporally and spatially resolved validation of emission inventories by measurements of ambient volatile organic compounds in Beijing, China, Atmos. Chem. Phys., 14, 5871-5891, 10.5194/acp-14-5871-2014, 2014.